# Plasma neurodegeneration biomarker concentrations associate with glymphatic and meningeal lymphatic measures in neurological disorders

Per Kristian Eide [1,2] ✉, Aslan Lashkarivand[1,2], Are Pripp[3,4], Lars Magnus Valnes[1], Markus Herberg Hovd [5], Geir Ringstad [6,7], Kaj Blennow [8,9] & Henrik Zetterberg [8,9,10,11,12,13]

Clearance of neurotoxic brain proteins via cerebrospinal fluid (CSF) to blood has recently emerged to be crucial, and plasma biomarkers of neurodegeneration were newly introduced to predict neurological disease. This study examines in 106 individuals with neurological disorders associations between plasma biomarkers [40 and 42 amino acid-long amyloid-β (Aβ40 and Aβ42), total-tau, glial fibrillary acidic protein (GFAP), and neurofilament light (NfL)] and magnetic resonance imaging measures of CSF-mediated clearance from brain via extra-vascular pathways (proxy of glymphatic function) and CSF-to-blood clearance variables from pharmacokinetic modeling (proxy of meningeal lymphatic egress). We also examine how biomarkers vary during daytime and associate with subjective sleep quality. Plasma concentrations of neurodegeneration markers associate with indices of glymphatic and meningeal lymphatic functions in individual- and disease-specific manners, vary during daytime, but are unaffected by sleep quality. The results suggest that plasma concentrations of neurodegeneration biomarkers associate with measures of glymphatic and meningeal lymphatic function.

Abnormal aggregations in the brain of toxic proteins such as amyloid-β (Aβ) and phosphorylated tau (P-tau), the main constituents of amyloid plaques and neurofibrillary tangles, are key features in neurodegeneration diseases. Several different mechanisms have been proposed as important for promoting aggregation, such as increased production or

decreased clearance, sequence variants increasing hydrophobicity, protein truncations or post-translational modifications, and chaperone proteins[1]. Clearance of brain proteins occurs along different transport routes, such as passage across the blood–brain-barrier (BBB) to the systemic circulation, cellular degradation of metabolites in the

[1]Dept. of Neurosurgery, Oslo University Hospital-Rikshospitalet, Oslo, Norway. [2]Institute of Clinical Medicine, Faculty of Medicine, University of Oslo, Oslo, Norway. [3]Oslo Centre of Biostatistics and Epidemiology, Research Support Services, Oslo University Hospital, Oslo, Norway. [4]Faculty of Health Sciences, Oslo Metropolitan University, Oslo, Norway. [5]Section for Pharmacology and Pharmaceutical Biosciences, Department of Pharmacy, University of Oslo, Oslo, Norway. [6]Dept. of Radiology, Oslo University Hospital—Rikshospitalet, Oslo, Norway. [7]Department of Geriatrics and Internal medicine, Sorlandet Hospital, Arendal, Norway. [8]Department of Psychiatry and Neurochemistry, Institute of Neuroscience and Physiology, the Sahlgrenska Academy at the University of Gothenburg, Mölndal, Sweden. [9]Clinical Neurochemistry Laboratory, Sahlgrenska University Hospital, Mölndal, Sweden. [10]Department of Neurodegenerative Disease, UCL Institute of Neurology, Queen Square, London, UK. [11]UK Dementia Research Institute at UCL, London, UK. [12]Hong Kong Center for Neurodegenerative Diseases, Clear Water Bay, Hong Kong, China. [13]UW Department of Medicine, School of Medicine and Public Health, Madison, WI, USA. ✉e-mail: p.k.eide@medisin.uio.no

brain by macrophages and glia cells, and cerebrospinal fluid (CSF)-mediated clearance[2]. Independent of whether clearance occurs directly from the brain to blood or via lymph as an intermediate step, all metabolites eventually reach the blood circulation. Therefore, measurements of plasma concentrations of brain metabolites, such as 40 and 42 amino acid-long Aβ peptides (Aβ40 and −42), tau protein variants, glial fibrillary acidic protein (GFAP), and neurofilament light (NfL), have emerged as diagnostic biomarkers of neurodegenerative diseases, currently in particular for Alzheimer's disease (AD) and post-traumatic brain injuries[3,4]. At present, biomarkers like Aβ40, Aβ42, phosphorylated tau (P-tau), GFAP, and NfL make a huge impact on the diagnostics of neurodegenerative diseases[4].

The role of CSF-mediated clearance of brain metabolites has gained renewed interest during the last decade, not least facilitated by the discoveries of CSF-mediated molecular clearance from the brain via paravascular pathways (denoted the glymphatic system) in 2012[5], and of functional meningeal lymphatic vessels (enabling molecular egress from CSF to extracranial lymph) in 2015[6,7]; for review see also[2]. An increasing body of evidence points to the role of the glymphatic[5] and meningeal lymphatic systems[8] in the clearance of Aβ and tau. Importantly, the glymphatic and meningeal lymphatic systems seem to be under circadian control[9]. It is presently assumed that the glymphatic system is a waste clearance system mostly active during sleep[10], while the meningeal lymphatic system is active during the awake state[11]. Hence, under physiological circumstances, brain metabolites may be mobilized from the brain to CSF and lymphatic structures during nighttime, while clearance via the meningeal lymphatic system is enhanced during daytime. From this, it might be expected that plasma concentrations of CNS-derived metabolites and biomarkers for neurodegenerative disease processes are impacted in concert by the glymphatic and meningeal lymphatic systems. However, it remains unclear how alterations in CSF-mediated clearance of CNS-derived molecules via the glymphatic and meningeal lymphatic systems relate to their plasma concentrations. These mechanisms are important to understand, not least because potentially modifiable clearance routes may represent a target for treating disease processes underlying changes in plasma biomarker concentrations.

This study was undertaken to examine in humans how plasma concentrations of neurodegeneration biomarkers associate with tentative indices of glymphatic and meningeal lymphatic functions. The glymphatic function was examined as the brain-wide distribution of a magnetic resonance imaging (MRI) contrast agent, gadobutrol, administered intrathecal, and utilized as a CSF tracer[12]. The meningeal lymphatic function was examined by the CSF-to-blood clearance recently described by our group[13]. The MRI contrast agent gadobutrol is hydrophilic and distributes freely in CSF, and enriches the extravascular spaces of the brain[12]. We aimed to address the following hypotheses: (1) Whether plasma concentrations of CNS-derived

biomarkers correlate with indices of glymphatic clearance, assessed by extravascular brain enrichment of an intrathecal CSF tracer. The tracer does not cross the intact BBB and can here thus serve as a surrogate marker of clearance along extravascular pathways. Recent research suggests that impaired extravascular clearance (glymphatic function) is a major cause of deteriorated clearance of metabolites from the brain to CSF and further to meningeal lymphatic transport routes[5,12]. A negative correlation might be expected, provided that increased tracer levels in the brain during the clearance phase (24 and 48 h) reflects impaired glymphatic function. (2) Whether plasma biomarker concentrations correlate with indices of meningeal lymphatic function, assessed by variables of CSF-to-blood clearance. We recently reported a population pharmacokinetic model for CSF-to-blood clearance of the presently used CSF tracer (gadobutrol)[14]. The hypothesis is that meningeal lymphatic clearance represents a final common pathway for the egress of metabolites from CSF[15]. Hypothetically, in subjects with intact BBB and no abnormal CSF leakage, the CSF-to-blood clearance may reflect the total meningeal lymphatic clearance capacity. In line with this hypothesis, we previously demonstrated enrichment in the parasagittal dura of this intrathecal CSF tracer[16]. Furthermore, the CSF-to-blood clearance variables of the pharmacokinetic model showed large variation between individuals and across disease catgeories[14]. Therefore, the direction of the correlation might be expected to vary. (3) Whether plasma biomarker concentrations vary during daytime and relate to chronically impaired sleep quality. This could be expected given the proposed role of sleep and circadian rhythm on both glymphatic and meningeal lymphatic clearance function[9,10,17,18].

## Results
### Study cohort
The present study included 106 patients who underwent intrathecal contrast-enhanced MRI as part of a work-up for tentative CSF disturbances (Table 1). Fifty-four of the patients were iNPH subjects, who fulfilled the criteria of "Definite" iNPH, according to the Japanese guidelines[19]. The other cohorts consisted of patients who were diagnosed with communicating hydrocephalus (cHC), arachnoid cyst (AC), idiopathic intracranial hypertension (IIH), or spontaneous intracranial hypotension (SIH), and who underwent surgery for these diseases. Twelve individuals were denoted reference (REF) subjects in which no apparent CSF disturbance was diagnosed. As compared with the REF subjects, the patient cohorts with iNPH and arachnoid cysts (ACs) were older and with different sex distributions.

### Association between plasma concentrations of neurodegeneration biomarkers and CSF tracer clearance from the brain (proxy of glymphatic function)
Figure 1 shows the tracer enrichment after 24 h in the iNPH cohort, as an indicator of glymphatic function. Figure 1A shows the average

## Table 1 | Patient material

| Variable | Total material | Patient subgroups | | | | | |
|---|---|---|---|---|---|---|---|
| | | REF | iNPH | cHC | AC | IIH | SIH |
| **N** | 106 | 12 | 54 | 8 | 9 | 13 | 10 |
| **Demographic** | | | | | | | |
| Gender (F/M) | 56/50 | 11/1 | 21/33[c] | 3/5[a] | 4/5[a] | 11/2 | 6/4 |
| Age (yrs) | 58.7 ± 18.1 | 38.4 ± 13.9 | 72.5 ± 5.7[c] | 51.8 ± 11.4 | 55.9 ± 18.8[b] | 34.1 ± 11.8 | 48.5 ± 8.4 |
| BMI (kg/m²) | 27.6 ± 5.0 | 27.8 ± 5.2 | 27.2 ± 4.4 | 24.9 ± 4.0 | 26.4 ± 3.5 | 32.3 ± 4.9 | 26.7 ± 6.6 |
| **Subjective sleep quality** | | | | | | | |
| PSQI score (total) | 9.2 ± 4.4 | 8.1 ± 4.9 | 11.5 ± 2.1 | 6.7 ± 3.9 | 8.9 ± 4.5 | 11.9 ± 4.2 | 8.7 ± 3.2 |

Continuous data presented as mean ± standard deviation.
Differences towards REF group indicated by: [a]$P < 0.05$, [b]$P < 0.01$, [c]$P < 0.001$; analysis of variance (ANOVA) with Bonferroni post hoc tests for continuous data and Pearson Chi-square test for categorical data. Source data are provided as Source Data file.

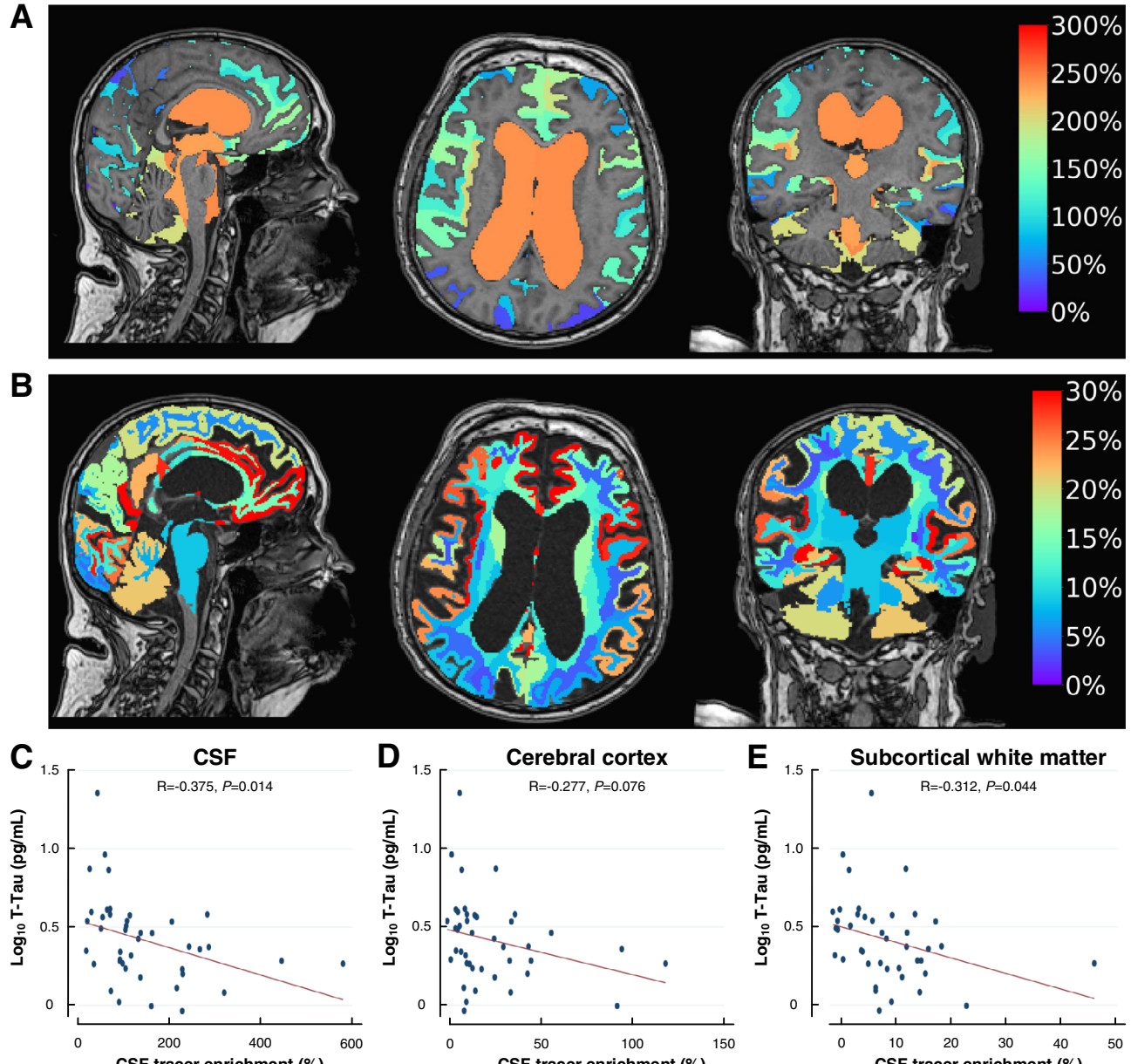

**Fig. 1 | Association between clearance of a CSF tracer and plasma concentrations of log$_{10}$ T-tau in the iNPH cohort ($n$ = 54). A** The percentage changes in tracer within CSF (ventricular and subarachnoid spaces) of the iNPH cohort at 24 h compared to Pre are shown in sagittal, axial and coronal planes. Higher values indicate reduced clearance. **B** The percentage change in tracer enrichment within the brain (tracer in CSF subtracted) of the iNPH cohort at 24 h is shown in the sagittal, axial, and coronal planes. The panels (**A, B**) show the average change in tracer enrichment at 24 h at the group level. In lower panels (**C–E**) are shown correlations between average plasma concentrations of log$_{10}$ T-tau (log$_{10}$ T-tau) in plasma and change in tracer enrichment at 24 h within **C** CSF, **D** cerebral cortex, and **E** subcortical white matter. Each panel **C–E** shows the scatter plot with Pearson correlation coefficient, fit line, and significance level. A higher percentage change in tracer enrichment (i.e., normalized T1 signal) at 24 h reflects stronger tracer enrichment and impaired clearance of tracer (i.e., impaired glymphatic function). This was associated with reduced plasma T-tau concentration. Source data are provided as Source Data file.

change in tracer enrichment within the ventricular and subarachnoid CSF spaces, and Fig. 1B the change in tracer enrichment within the brain. Within the iNPH group, there was a significant negative correlation between plasma T-tau concentration and tracer enrichment in CSF, cerebral cortex, and subcortical white matter (Fig. 1C–E), suggesting that impaired glymphatic function (i.e., the higher percentage change in CSF tracer enrichment) associates with reduced plasma T-tau concentration. For tau brain clearance, the extravascular clearance route via CSF and meningeal lymphatics to blood may be significant. It was previously reported that tau lacks a dedicated BBB-transporter[2]. However, some evidence also suggests the passage of tau

via the BBB[20], indicating that the brain also contributes a proportion of plasma T-tau concentrations.

Figure 2A–F shows associations between plasma concentrations of neurodegeneration biomarkers and tracer enrichment (indicative of glymphatic function) in the cerebral cortex and subcortical white matter for the entire study group of 106 subjects. There were significant negative correlations between tracer enrichment in the cerebral cortex (indicative of glymphatic clearance from the cerebral cortex) and plasma concentrations of T-tau (48 h; Fig. 2A), GFAP (24 and 48 h; Fig. 2B) and NfL (24 and 48 h; Fig. 2C). There were also significant negative correlations between tracer enrichment in

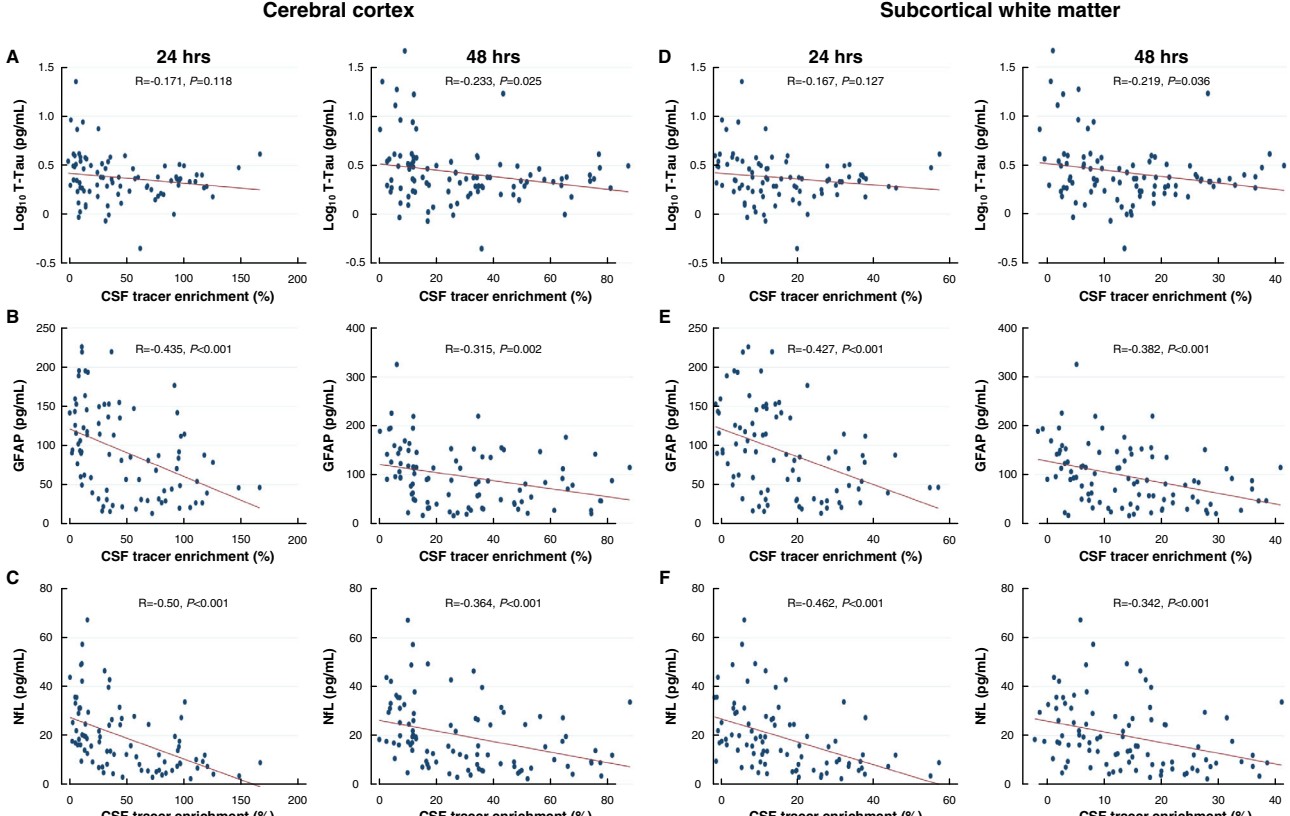

**Fig. 2 | Association between plasma concentrations of neurodegeneration biomarkers and enrichment of CSF tracer within the cerebral cortex and subcortical white matter of the entire cohort ($n = 106$).** Plots of associations between tracer enrichment in the cerebral cortex at 24 h and average plasma concentrations of **A** T-tau ($log_{10}$ T-tau), **B** GFAP, and **C** NfL. Plots for associations between tracer enrichment in subcortical white matter at 24 and 48 h and average plasma concentrations of **D** T-tau, **E** GFAP, and **F** NfL Each scatter plot shows the linear fit line, including Pearson correlations and significance levels. Source data are provided as Source Data file.

subcortical white matter (indicative of glymphatic clearance from the subcortical white matter) and plasma concentrations of T-tau (48 h; Fig. 2D), GFAP (24 and 48 h; Fig. 2E) and plasma NfL concentration (24 and 48 h; Fig. 2F). Accordingly, higher CSF tracer enrichment in the cerebral cortex and subcortical white matter at 24 and 48 h (clearance phase), indicative of impaired glymphatic function, were accompanied with lower plasma biomarker concentrations.

Of note, there was a negative correlation between clearance of tracer from CSF and plasma Aβ42/Aβ40 ratio that was significant in the iNPH cohort at 24 h (Supplementary Fig. 1A, B) and the total cohort at 48 h (Supplementary Fig. 1C, D). Accordingly, reduced clearance from CSF seemed to be associated with a relatively reduced clearance to the blood of Aβ42.

### Association between plasma biomarker concentrations and pharmacokinetic model-based CSF-to-blood clearance variables (proxy of meningeal lymphatic efflux)

The CSF-to-blood clearance parameters of the presently included patients were calculated using a previously described population pharmacokinetic model[14]. Figure 3A–L shows the marked variation in CSF-to-blood clearance capacity between individuals and across patient groups. Table 2 presents the average pharmacokinetic variables for the entire cohort and the individual patient subgroups.

When addressing the iNPH cohort specifically, there was a significant negative correlation between lag-time and plasma Aβ42 (Fig. 3M), which was also seen for the entire group (Fig. 3N). Increased lag-time is indicative of the tracer taking longer to reach the site of clearance in the brain, or the tracer remaining longer in CSF before CSF-to-blood clearance is initiated.

The finding that CSF-to-blood clearance variables are highly individual (Fig. 3A–L) is reflected in the large variability of correlations between plasma biomarker concentrations and CSF-to-blood clearance variables (Fig. 4). Relationships were different across disease categories, with correlations being both positive and negative for the various categories. These observations are highly significant as they highlight the need for assessments at the single subject level since underlying biological processes such as CSF-to-blood clearance capacity are highly individual[14].

Moreover, when considering the total cohort, the correlation between CSF-to-blood clearance variables and plasma biomarker concentrations, the direction and strength of correlation varied among the biomarkers (Supplementary Fig. 2). Plasma GFAP concentration in the total cohort showed a significant positive correlation with absorption half-life ($T_{1/2, abs}$), time to maximum concentration ($T_{max}$), and lag-time ($T_{lag}$) (Supplementary Fig. 3), suggesting that the variables indicative of impaired CSF-to-blood clearance were associated with higher plasma GFAP concentration.

### Daytime variation in plasma biomarker concentrations

In line with the hypothesis, repeated blood samples revealed daytime variation in the plasma concentrations of several biomarkers. For the entire cohort, plasma concentrations increased during the daytime for T-tau, GFAP, and NfL (Fig. 5). Notably, the daytime variation differed between the different patient categories. Within the iNPH cohort, the plasma concentrations of Aβ declined significantly towards the evening, while T-tau and NfL concentrations increased significantly during the day (Fig. 6A–F). For the IIH cohort, both Aβ40 and Aβ42 concentrations declined significantly during the day (with no change in

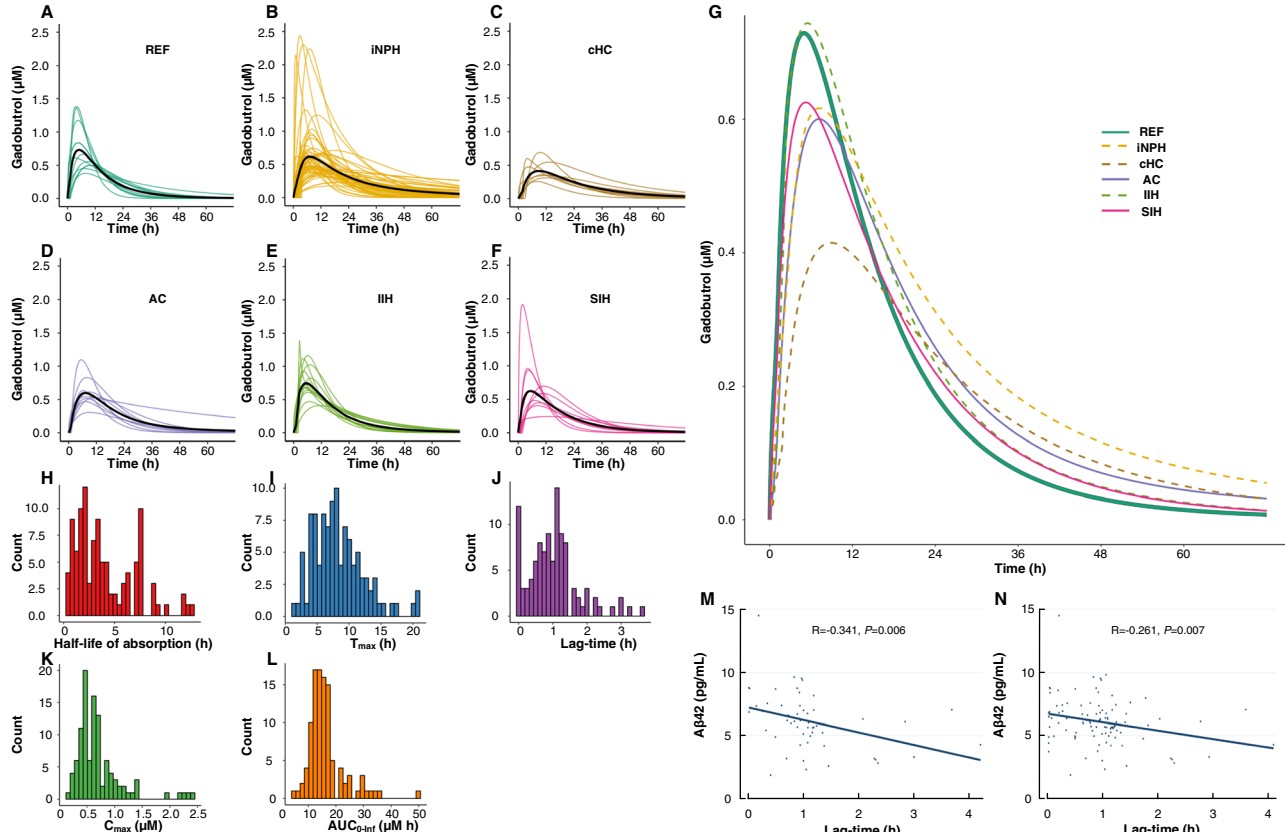

**Fig. 3 | CSF-to-blood clearance.** Inter-individual variability in CSF-to-blood clearance for the diagnosis categories shown for the **A** reference (REF), **B** idiopathic normal pressure hydrocephalus (iNPH), **C** communicating hydrocephalus (cHC), **D** arachnoid cyst (AC), **E** idiopathic intracranial hypertension (IIH), and **F** spontaneous intracranial hypotension group. The curves show individual posterior dose-normalized predicted concentrations of gadobutrol over time, and the black lines signify the mean concentration for each group, averaged at each time point. **G** The individual posterior predicted dose-normalized blood concentrations of intrathecal gadobutrol from the population pharmacokinetic model, averaged at each time point by group, illustrates differences between groups. The distribution of individual pharmacokinetic parameters for the entire cohort is shown as a histogram of parameter distribution for the **H** absorption half-life ($T_{1/2, abs}$), **I** time to maximum concentration ($T_{max}$), **J** lag-time ($T_{lag}$), **K** maximum concentration ($C_{max}$), and **L** area under the curve (AUC) from zero to infinity for the entire cohort of patients ($n = 106$). The association between pharmacokinetic parameters and plasma biomarker concentrations is here illustrated by a significant negative correlation between average plasma concentrations of Aβ42 and lag-time ($T_{lag}$) in **M**, the iNPH cohort, and **N**, the entire patient cohort. Scatter plots are shown with fit lines, Pearson correlation coefficients, and significance levels. Source data are provided as Source Data file.

**Table 2 | CSF-to-blood clearance variables for the entire cohort and the different patient categories**

| Variable | Total material | Patient subgroups | | | | | |
|---|---|---|---|---|---|---|---|
| | | REF | iNPH | cHC | AC | IIH | SIH |
| **CSF-to-blood clearance** | | | | | | | |
| Absorption half-life ($T_{1/2, abs}$) | 3.9 ± 2.5 | 3.3 ± 2.5 | 4.2 ± 2.5 | 4.7 ± 2.6 | 4.3 ± 2.6 | 2.5 ± 1.9 | 3.6 ± 2.9 |
| Lag-time ($T_{lag}$) | 1.0 ± 0.7 | 0.5 ± 0.5 | 1.1 ± 0.8[b] | 1.0 ± 0.5 | 1.0 ± 0.5 | 0.6 ± 0.5 | 0.9 ± 0.6 |
| Area under curve (AUC) | 56.8 ± 28.7 | 57.8 ± 17.7 | 51.7 ± 30.7 | 55.0 ± 33.2 | 66.6 ± 36.4 | 73.3 ± 17.2 | 54.6 ± 24.8 |
| Maximum concentration ($C_{max}$) | 2.7 ± 2.0 | 3.5 ± 2.1 | 2.2 ± 1.9[a] | 1.7 ± 0.7[a] | 2.7 ± 1.3 | 3.9 ± 1.5 | 3.1 ± 2.8 |
| Time to maximum concentration ($T_{max}$) | 8.1 ± 4.0 | 6.0 ± 2.5 | 9.1 ± 4.6[a] | 9.4 ± 3.1 | 7.9 ± 1.9 | 5.7 ± 2.1 | 7.5 ± 3.9 |
| **Renal clearance** | | | | | | | |
| GFR (ml/min/1,73m2) | 87.1 ± 18.6 | 102.5 ± 11.8 | 77.0 ± 14.4 | 90.4 ± 16.9 | 84.3 ± 20.0 | 104.5 ± 16.2 | 101.8 ± 10.3 |

Data presented as mean ± SD: Differences towards REF group indicated by: [a]$P < 0.05$, [b]$P < 0.01$ (regression analysis). Source data are provided as Source Data file.

Aβ42/Aβ40 ratio), while GFAP concentration declined towards evening (Fig. 6G–L). No daytime variation was observed for the other subgroups, except for a significant reduction in T-tau concentration towards evening in the SIH cohort. Therefore, the daytime variation in plasma biomarker concentrations seems to depend on the underlying disease.

## Association between plasma biomarker concentrations and subjective sleep quality

In the patient cohort presented here, we found no significant correlations between subjective sleep quality, measured by the total score of the Pittsburg sleep quality index (PSQI), and plasma biomarker concentrations (Supplementary Fig. 4). Furthermore, the good (PSQI

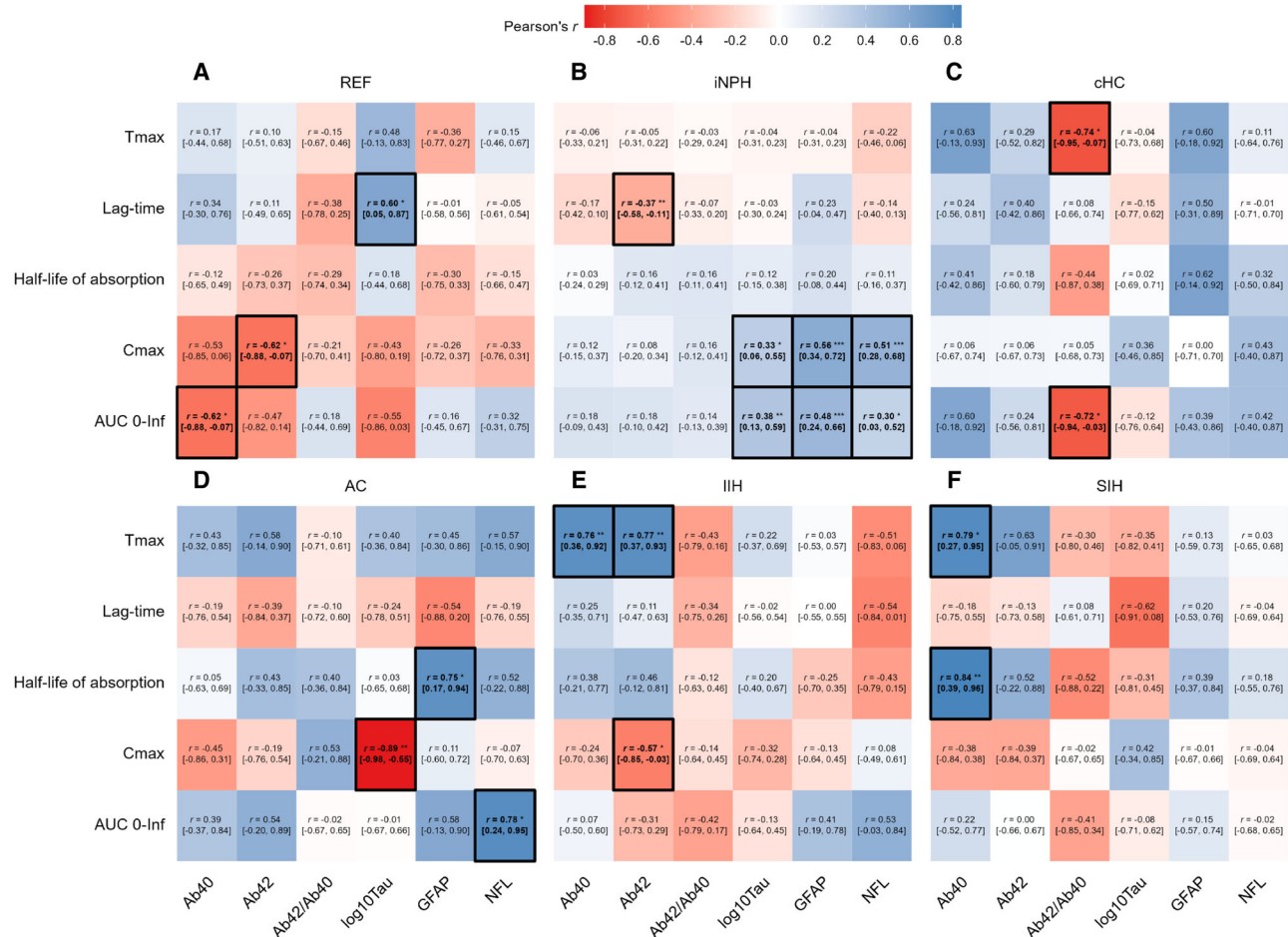

**Fig. 4 | Associations between CSF-to-blood clearance parameters and biomarker concentrations for the different disease categories.** The associations between CSF-to-blood clearance pharmacokinetic variables and plasma biomarker concentrations are shown as Pearson's correlations (*r*) [95% confidence interval] for the CSF disease categories **A** reference (REF), **B** idiopathic normal pressure hydrocephalus (iNPH), **C** communicating hydrocephalus (cHC), **D** arachnoid cyst (AC), **E** idiopathic intracranial hypertension (IIH), and **F** spontaneous intracranial hypotension (SIH). Significance levels of Pearson correlation coefficients: *$P < 0.05$, **$P < 0.01$, ***$P < 0.001$ (values in bold are statistically significant to $P < 0.05$); red boxes for negative correlations and blue boxes for positive correlations. Source data are provided as Source Data file.

total score ≤5) and poor (PSQI total score >6) sleepers showed no differences in plasma biomarker concentrations (Supplementary Table 1).

### Group differences in plasma biomarker concentrations

Significant differences were found between the iNPH and REF subjects for T-tau, GFAP, and NfL, revealed as higher concentrations of T-tau, GFAP, and NfL in the iNPH cohort (Fig. 7). The group differences during morning and evening are further shown in Table 3. After correction for group differences in renal function, i.e., glomerular filtration rate (GFR), the plasma GFAP and NfL concentration differences remained significant. We may, however, not conclude to which degree the observed differences between REF and iNPH subjects are disease-specific or were caused by age differences, as the iNPH cohort was about three decades older than the REF cohort. After correction for age, the group differences were not significant. On the other hand, age is a key factor behind iNPH.

## Discussion

This study provides several lines of evidence suggesting that plasma concentrations of neurodegeneration biomarkers associate with indices of glymphatic and meningeal lymphatic functions: (1) There was a significant correlation between levels of an intrathecal CSF tracer in the brain during the clearance phase (24

and 48 h; indicative of glymphatic function) and plasma biomarker concentrations. Hence, for the entire cohort, impaired tracer clearance from the brain correlated significantly with lower plasma concentrations of T-tau (Log₁₀ T-tau), GFAP, and NfL, while Aβ40 and Aβ42 did not show this pattern. (2) There were significant correlations between CSF-to-blood clearance variables (indicative of meningeal lymphatic capacity) and plasma biomarker concentrations, but the relationships were biomarker- and disease-specific. This may be related to the extensive inter-individual variation in CSF-to-blood clearance capacity. (3) In the total cohort, plasma T-tau, GFAP, and NfL concentrations changed significantly during the day, while we found no daytime variability in plasma concentrations of Aβ40 and Aβ42. In this cohort, subjective sleep quality was not associated with plasma biomarker concentrations. (4) Plasma GFAP, NfL and T-tau concentrations were increased in iNPH compared with REF subjects, but the difference could be caused by age differences.

We here examined the association between plasma concentrations of neurodegeneration biomarkers and indices of glymphatic and meningeal functions in patient groups that may not necessarily be categorized as neurodegenerative diseases. In this regard, it should be noted that the current knowledge about the plasma biomarkers referred to here, primarily derives from studies of Alzheimer's disease, frontotemporal dementia, atypical parkinsonian disorders,

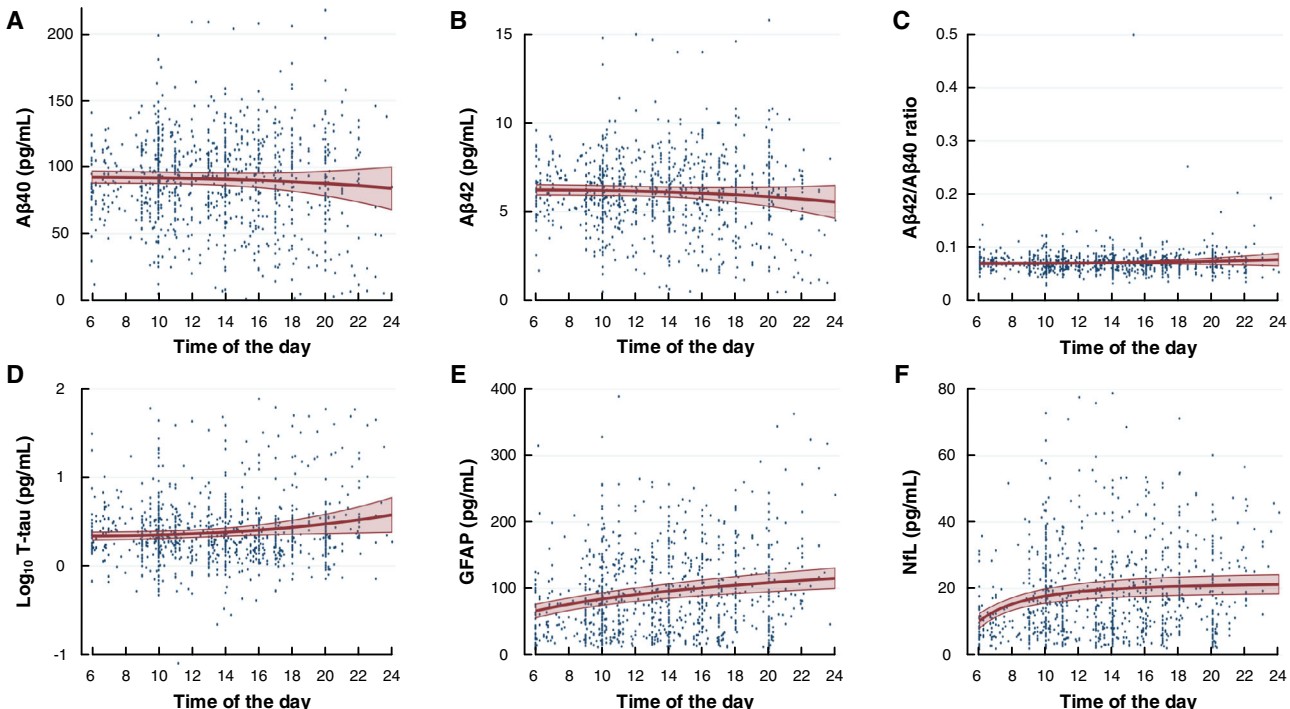

**Fig. 5 | Daytime variation in plasma biomarker concentrations within the entire cohort ($n = 106$).** Biomarker concentrations in longitudinally collected plasma samples showed no difference over time for **A** Aβ40 ($P = 0.348$), **B** Aβ42 ($P = 0.185$), or **C** Aβ42/Aβ40 ratio ($P = 0.309$). Increasing plasma concentrations toward evening was shown for **D** Log₁₀ T-tau ($P = 0.031$), **E** GFAP ($P < 0.001$), and **F** NfL ($P < 0.001$; fractional polynomial regression analysis with robust standard error for repeated measurements). Results presented as mean (red line) with 95% confidence intervals (red shaded area), including individual data points. Source data are provided as Source Data file.

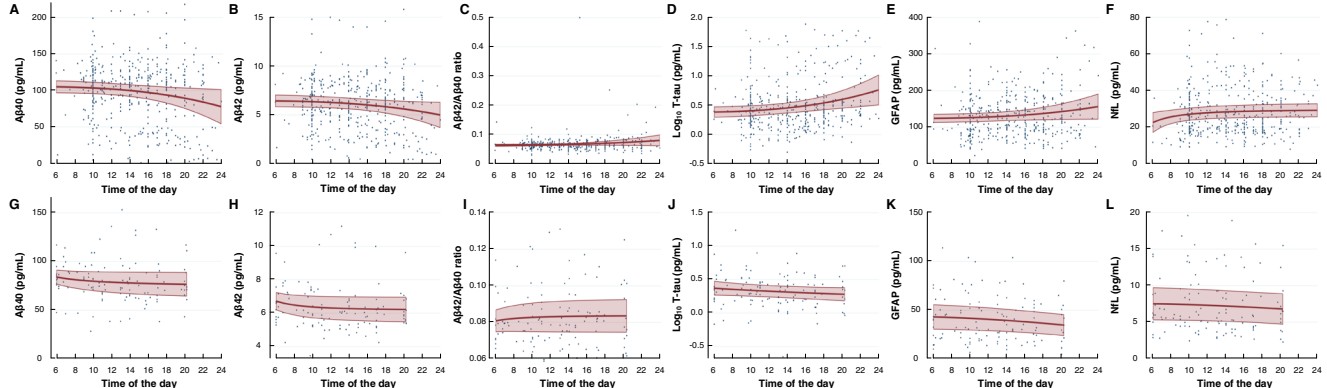

**Fig. 6 | Daytime variation in plasma biomarker concentrations within the iNPH ($n = 54$) and IIH ($n = 13$) cohorts.** Repeated measurements of biomarker concentrations in plasma of iNPH showed different profiles depending on the biomarker examined: **A** Aβ40 ($P = 0.039$), **B** Aβ42 ($P = 0.055$), **C** Aβ42/Aβ40 ratio ($P = 0.134$), **D** T-tau ($P < 0.001$), **E** GFAP ($P = 0.072$), and **F** NfL ($P = 0.030$; regression analysis). The profile of daytime variation for IIH subjects differed: **G** Aβ40

($P = 0.0049$), **H** Aβ42 ($P = 0.035$), **I** Aβ42/Aβ40 ratio ($P = 0.304$), **J** T-tau ($P = 0.130$), **K** GFAP ($P = 0.026$), and **L** NfL ($P = 0.193$; fractional polynomial regression analysis with robust standard error for repeated measurements). Results presented as mean (red line) with 95% confidence intervals (red shaded area), including individual data points. Source data are provided as Source Data file.

and post-traumatic brain injury[4]. The existing knowledge about the present biomarkers are shortly commented on. In AD, extracellular Aβ aggregation is a key pathogenic factor, and measurements of Aβ40 and Aβ42 in CSF and plasma are currently applied biomarkers with the ability to show abnormality long before clinical disease manifestation. Since the Aβ42 isoform is more prone to aggregation within the brain than Aβ40, while the latter isoform is most abundant, a reduced CSF Aβ42/Aβ40 ratio is a robust marker of AD pathology[4,21]. The role of plasma Aβ42/Aβ40 ratio as a marker of Aβ pathology has been debated[22], though recent data indicate the plasma Aβ42/Aβ40 ratio

indeed reflects Aβ pathology in the brain, although not as robustly as CSF Aβ42/Aβ40[23]. Human in vivo evidence supports the assumption that impaired clearance from CSF is associated with Aβ deposition and Aβ pathology in the brain of AD subjects[24]. With regard to the tau protein, intra-neuronal accumulation of hyperphosphorylated tau in full-length or tangled forms characterizes AD and other tauopathies. Both CSF and blood biomarkers of phosphorylated tau protein (P-tau) have been introduced to assess tau pathology[25]. Available plasma tau assays address phosphorylation at different amino acids, such as 181 (P-tau-181), 217 (P-tau-217), and 231 (P-tau 231)[26]. Increased plasma

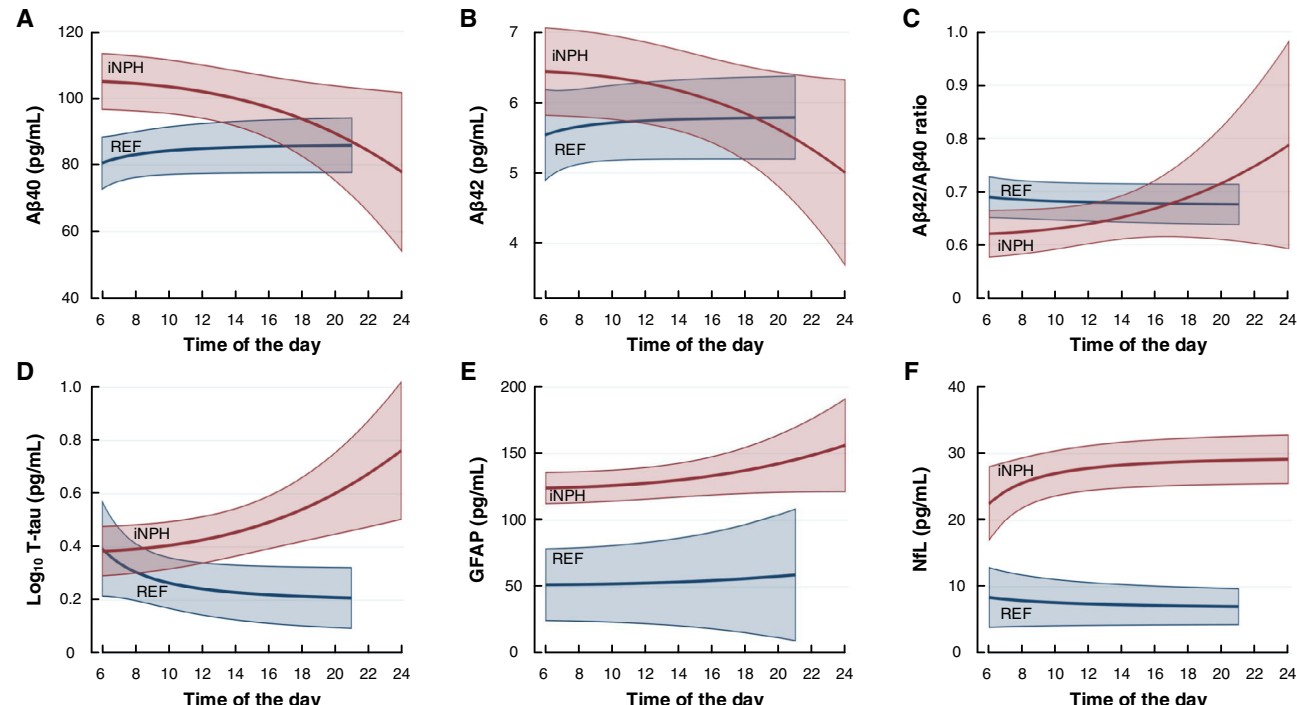

**Fig. 7 | Daytime differences in plasma biomarker concentrations between the iNPH and REF categories.** Biomarker concentrations in longitudinally collected plasma samples were compared between iNPH and reference subjects for **A** Aβ40 ($P = 0.106$), **B** Aβ42 ($P = 0.431$), or **C** Aβ42/Aβ40 ratio ($P = 0.835$), **D** T-tau (log$_{10}$ Tau; $P = 0.012$), **E** GFAP ($P < 0.001$), and **F** NfL ($P < 0.001$; regression analysis of average plasma biomarker concentrations). Results presented as mean (red line for iNPH and blue line for REF) with 95% confidence intervals (red shaded area for iNPH and blue shaded area for REF). Source data are provided as Source Data file.

## Table 3 | Plasma concentrations of neurodegeneration biomarkers in the different patient cohorts

| | REF | iNPH | cHC | AC | IIH | SIH |
|---|---|---|---|---|---|---|
| **Aβ40 (pg/ml)** | | | | | | |
| 06:00 – 12:00 | 81.1 ± 13.0 | 102.2 ± 31.3[a] | 81.8 ± 14.3 | 89.5 ± 28.2 | 79.6 ± 16.8 | 83.8 ± 18.3 |
| 12:00 – 24:00 | 85.4 ± 16.0 | 98.1 ± 39.6 | 80.5 ± 27.0 | 90.2 ± 27.2 | 78.9 ± 21.7 | 88.9 ± 22.4 |
| **Aβ42 (pg/ml)** | | | | | | |
| 06:00 – 12:00 | 5.5 ± 0.9 | 6.2 ± 2.1 | 5.7 ± 1.0 | 6.3 ± 1.5 | 6.4 ± 1.0 | 6.0 ± 1.1 |
| 12:00 – 24:00 | 5.8 ± 1.2 | 6.1 ± 2.6 | 5.6 ± 1.4 | 6.3 ± 1.4 | 6.3 ± 1.3 | 6.6 ± 1.4 |
| **Aβ42/Aβ40 ratio** | | | | | | |
| 06:00 – 12:00 | 0.068 ± 0.006 | 0.063 ± 0.013 | 0.070 ± 0.009 | 0.074 ± 0.015 | 0.083 ± 0.015[b] | 0.074 ± 0.012 |
| 12:00 – 24:00 | 0.068 ± 0.006 | 0.068 ± 0.024 | 0.073 ± 0.013 | 0.073 ± 0.013 | 0.082 ± 0.016 | 0.076 ± 0.014 |
| **Log$_{10}$ T-tau** | | | | | | |
| 06:00 – 12:00 | 0.27 ± 0.29 | 0.44 ± 0.33 | 0.26 ± 0.18 | 0.42 ± 0.24 | 0.36 ± 0.19 | 0.35 ± 0.13 |
| 12:00 – 24:00 | 0.21 ± 0.18 | 0.48 ± 0.39[b] | 0.29 ± 0.42 | 0.31 ± 0.18 | 0.28 ± 0.18 | 0.32 ± 0.11 |
| **GFAP** | | | | | | |
| 06:00 – 12:00 | 53.1 ± 53.4 | 123.9 ± 57.1[c] | 85.7 ± 26.8 | 78.5 ± 54.3 | 42.4 ± 23.6 | 45.6 ± 26.7 |
| 12:00 – 24:00 | 52.2 ± 58.4 | 134.6 ± 52.4[c] | 86.9 ± 28.7 | 76.0 ± 57.2 | 38.0 ± 19.8 | 47.0 ± 24.2 |
| **NfL** | | | | | | |
| 06:00 – 12:00 | 7.8 ± 6.2 | 27.9 ± 12.8[c] | 12.0 ± 4.4 | 19.5 ± 10.9 | 7.6 ± 4.0 | 11.1 ± 3.9 |
| 12:00 – 24:00 | 7.2 ± 5.5 | 27.7 ± 12.0[c] | 11.4 ± 4.3 | 16.8 ± 9.3 | 7.0 ± 3.7 | 11.0 ± 3.7 |

Data presented as mean ± SD: Differences towards REF group indicated by: [a]$P < 0.05$, [b]$P < 0.01$, [c]$P < 0.001$. Source data are provided as Source Data file.

P-tau isoform concentration relates to the degree of cerebral Aβ pathology, predicts the density of tau tangles, and differentiates AD from other dementias[27,28]. At present, P-tau is primarily considered a marker of AD, and not for other types of dementia. The T-tau presented in the present study does not refer to any particular type of phosphorylation and is not specific to AD. Notably, the cerebral aggregations of Aβ and P-tau associate with inflammatory responses and neuronal and glial damage. Here we focused on GFAP and NfL. GFAP is a marker of astrocyte activation, not specific for AD, but as well for traumatic brain injury and a variety of other brain diseases[29]. In the context of AD pathology, GFAP is considered to reflect an astrocyte response to Aβ aggregation in the brain[30,31], but not tau pathology[31].

Notably, the association between GFAP and Aβ pathology is stronger for plasma than CSF pathology[30,31]. NfL is a non-specific marker of neuroaxonal injury and is found to increase in a variety of neurodegenerative disorders[32], with increasing levels during aging[33]. The CSF and blood concentrations of NfL correlate[34]. One study reported increased CSF and plasma concentrations of NfL 7 years before symptom onset in autosomal dominant AD subjects who were cognitively intact[35].

While the present observations suggest an association between concentrations of plasma biomarkers of neurodegeneration disease and indices of glymphatic and meningeal lymphatic clearance function, we remind that the specific efflux routes from the brain and CSF to the blood of the present biomarkers are not fully understood. Additional pathways than via CSF are involved[2]. To this end, the passage route for Aβ has been most studied; for which passage via BBB is important, as well as cellular degradation within the brain by macrophages and microglia, in addition to egress via CSF[2]. The latter may perhaps be more important than previously considered, given recent data on an important clearance route of Aβ via meningeal lymphatic vessels and the beneficial effect of lymphatic enhancement[8]. The recent observations that Aβ-labeled cells were far more abundant in cervical than inguinal lymph nodes support this view[36]. Perhaps enhancement of meningeal lymphatic function could be one way to modify CSF-mediated Aβ clearance?. The transport route for the tau protein, on the other hand, is less characterized; no dedicated BBB transport has been identified[2], but passage across the BBB was observed in mice[20]. Moreover, experimental data indicate that efflux via dural meningeal pathways may be crucial[37]. The efflux routes for GFAP and NfL are largely unknown. In this regard, it should be emphasized that neurodegenerative diseases involve BBB damage to a variable degree[38], which may facilitate the passage of CNS-derived biomarkers directly to the blood. With regard to the present observations, we do not expect BBB disruption to be a significant factor, though some impairment of BBB cannot be excluded. It is generally accepted that BBB becomes impaired with increasing age, which is relevant for the iNPH cohort with a median age of 72.5 ± 5.7 years. In addition, previous histopathological studies have shown some degree of BBB disruption in iNPH[39] and IIH[40] patients, though the effect on BBB is not major in these cases.

Here, we applied a surrogate marker for in vivo human assessment of glymphatic and meningeal lymphatic functions, as direct measurements cannot be done in humans. In vivo visualization of extra-BBB clearance pathways in humans can be done by multiphase and standardized MRI utilizing a CSF tracer[12]. However, given the low resolution of MRI (1 mm), the exact transport route of the CSF tracer at the microscopic level (paravascular vs interstitial transport) cannot be defined, though it may be concluded that tracer transport occurs extravascular. With the term "glymphatic", we refer to CSF-mediated tracer enrichment and clearance from the brain along extravascular pathways, well aware that this represents only one aspect of the glymphatic concept presented in 2012[5]. This concept as a whole is still heavily debated[41]; the controversy particularly concerns mechanisms behind the solute movement (role of convective forces versus diffusion), the role of aquaporin-4 (AQP4) located at perivascular astrocytic endfeet, the role of directionality of CSF movement, and lack of evidence for efflux along perivenous spaces. Despite this ongoing debate, we here use the term glymphatic with an awareness of these yet unresolved issues. In this regard, an obvious question is how glymphatic function assessed by a non-endogenous CSF tracer (gadobutrol) may be relevant for excretion from the brain and CSF of endogenous compounds such as Aβ40, Aβ42, T-tau, GFAP, and NfL? In soluble form, these proteins are expected to be transported paravascular in the human brain[42]. The CSF tracer used here, gadobutrol, has a molecular weight (MW) of 604 Da, which is many times smaller than the plasma metabolites: Aβ40 (MW about 4.3 kDa), Aβ42 (MW

about 4.5 kDa), tau (MW up to 80 kDa) GFAP (MW about 50 kDa), NfL (MW about 70 kDa). However, these large proteins may as well be transported paravascular. For example, one study demonstrated in pig brain glymphatic paravascular tracer enrichment of a CSF tracer [AlexaFluor647-conjugated bovine serum albumin (BSA-647)] with MW about 66 kDa[43]. The tracer enrichment in the pig brain utilizing this large MW compound was comparable to the CSF tracer (gadobutrol) enrichment seen in the human brain after intrathecal injection[43]. Second, gadobutrol is a hydrophilic substance that does not pass the intact BBB, nor does it permeate the cell membrane or interact with other molecules of the brain. Third, the transport of plasma biomarkers and CSF tracer in the subarachnoid compartments may differ due to the membranes compartmentalizing the subarachnoid space[44,45]. In this regard, ref. 45 recently described a subarachnoid lymphatic-like membrane (SLYM) compartmentalizing the subarachnoid space, where this mesothelial membrane was impermeable to substances >3 kDa.

With regard to meningeal lymphatic clearance capacity, no specific methods for assessments have been established. Verma et al.[46] used technetium-99 m (99mTc-) diethylenetriaminepentaacetic acid (DTPA) imaging to estimate clearance from CSF to the periphery after intrathecal injection. This method as well provides for visualization of tracer. It may be considered a limitation that the radiolabel half-life of this compound is 6 h (time to maximum concentration in the present study was 8.1 ± 4.0 h). However, radiotracers may be used for dynamic scans that are three to four times longer than their half-life. Moreover, a recent study suggested the useful role of radiotracers for in vivo imaging of glymphatic function[47]. We recently introduced another method to assess CSF-to-blood clearance capacity[13], incorporating an intrathecal injection of a CSF tracer (e.g., gadobutrol), followed by repeated plasma samples. A population pharmacokinetic model based on consecutive blood samples ($n = 1140$) from 161 subjects demonstrated marked inter-individual variation and even marked variation across disease categories[14]. It is important to note that this method measures total CSF-to- blood clearance capacity, independent of the clearance route. For substances also passing the BBB, direct passage to blood may contribute. In subjects with impaired BBB integrity, which must be anticipated in older subjects[48], the passage of substances directly to blood may be expected to some degree. In other rare diseases, e.g. spontaneous intracranial hypotension due to CSF-to-venous-fistula[49], there is an abnormal direct passage to blood. However, for substances primarily being excreted via CSF, we tentatively suggest that the CSF-to-blood clearance capacity can be considered a surrogate marker of meningeal lymphatic clearance capacity, and that brain clearance may indirectly be dependent on meningeal lymphatic CSF clearance capacity. We previously reported that the intrathecal CSF tracer enriched extensively in parasagittal dura, but identified a few arachnoid granulations also being enriched by tracer[16]. The volume fraction of efflux via arachnoid granulations could not be determined, and requires further studies. However, knowledge of the exact clearance route is not critical, as the CSF-to-blood clearance aims to measure total clearance capacity, independent of the clearance pathway. For compounds largely excreted via CSF, the CSF-to-blood clearance capacity may largely reflect the total meningeal lymphatic clearance capacity.

It remains to be determined how CSF-to-blood clearance variables from pharmacokinetic modeling provide information about the underlying biological processes preceding neurodegenerative diseases. It may be of particular interest, however, that emerging evidence suggests that CSF-to-blood clearance by impaired meningeal function to be a risk factor for abnormal aggregation of toxic proteins (e.g., Aβ, tau, and α-synuclein)[7,8,37,50,51]. Failed CSF clearance has been shown to represent a feature of AD, related to Aβ deposition and to the pathology of AD[24]. Most importantly, CSF clearance failure is expected to occur in the preclinical stages of dementia disease[52], and impaired

CSF clearance due to failure of meningeal lymphatic function may be involved in other diseases, such as malignant glioma[53] and ischemic stroke[54]. Given the prospects of interventions that may enhance meningeal lymphatic function[8,55,56], direct estimation of CSF-to-blood clearance capacity, as described here, might be useful to monitor treatment regimens. Therefore, defining CSF-to-blood clearance changes in other diseases than dementias such as AD seems warranted.

The variable profiles of the presently reported plasma metabolites may reflect their different clearance routes from the brain. On the one hand, GFAP showed significant daytime variation, correlation with glymphatic function, and correlation with CSF-to-blood clearance, indicative of meningeal lymphatic function. On the other hand, Aβ40 and Aβ42 showed no daytime variation in the total cohort (though varied during the day in disease cohorts such as iNPH and IIH), no correlation with glymphatic function and no correlation with CSF-to-blood clearance for the entire cohort (though for certain disease categories). Moreover, both T-tau and NfL showed daytime variation in plasma concentrations and correlated with glymphatic function, while CSF-to-blood clearance was disease-specific. Tentatively, these observations may be interpreted as CSF clearance being more dominant for T-tau, GFAP, and NfL than for Aβ being excreted by different routes (Aβ is also less CNS-specific than the other proteins). In this regard, the observations of a closer association between Aβ pathology and plasma GFAP than what has been observed for CSF GFAP[30,31] might perhaps suggest direct release into the blood at the glia-neuro-vascular interface. The presently reported significant positive correlation between CSF-to-blood clearance and plasma GFAP, indicative of higher blood levels with impaired CSF-to-blood clearance, could support this assumption. If the correlation had been negative, it might rather indicate higher plasma levels due to more efficient CSF-to-blood clearance. The weaker association of CSF GFAP with Aβ pathology may, however, also reflect the recently discovered instability of the protein in CSF, which is much less of a problem in the plasma matrix[57]. Overall, the present results point to the need for individualized assessments concerning causes for altered plasma concentrations of neurodegeneration biomarkers.

Here, we addressed patient categories that has been less studied concerning plasma biomarkers of neurodegeneration disease. The existing knowledge about plasma biomarkers largely relates to AD, while their role in other brain disorders remains less understood. The patient category closest to AD is the iNPH cohort. The iNPH disease is a neurodegenerative disease and subtype of dementia with close histo-pathological overlap towards AD[58]; for example, accumulation in the brain of Aβ and/or tau is seen in a significant proportion of iNPH subjects[59]. Moreover, previous studies showed a comparable reduction of CSF production and turnover in AD and iNPH patients[60], though CSF Aβ40, Aβ42, and T-/P-tau also were able to differentiate iNPH from AD subjects[61]. iNPH involves impaired glymphatic function[12,62], which was proposed as a common mechanism behind dementias and AD[63]. The iNPH disease is also characterized by impaired CSF-to-blood clearance capacity[14]. In this study, impaired clearance of tracer from CSF, cerebral cortex, and subcortical white matter, indicative of impaired glymphatic function, was accompanied by reduced plasma T-tau concentration. This finding is of interest, considering the increased tau deposition within the brain of this patient group[59]. In line with this observation, iNPH patients not responding to shunt surgery had increased CSF T-tau concentration[64,65], and also increased CSF NfL concentration[65]. Cognitively impaired iNPH patients also had higher P-tau concentration than cognitively intact iNPH subjects[61]. They also presented a lower Aβ42/Aβ40 ratio[61]. Here, we found that impaired clearance of tracer from CSF was associated with a significantly lower plasma Aβ42/Aβ40 ratio in iNPH patients at 24 h and within the total cohort at 48 h after tracer injection. The present results also disclosed higher plasma concentrations of Aβ40, T-tau, GFAP, and NFL in the iNPH cohort, indicative of ongoing neurodegeneration. In comparison,

immunohistochemistry of brain tissue specimens of iNPH patients demonstrates astrogliosis with increased GFAP immunostaining[66]. It remains to be determined whether the reduced plasma concentrations of Aβ40 and Aβ42 and increased plasma levels of tau, GFAP, and NfL towards the evening reflect the clearance pathways and clearance rates of these proteins from the brain. A significant negative correlation in iNPH between plasma Aβ42 and lag-time could indicate that clearance via CSF and meningeal lymphatics has a major role in this isoform.

iNPH differed from the IIH disease, which is characterized by high intracranial pressure, impaired glymphatic function, and enhanced CSF-to-blood clearance[14,67]. In IIH, plasma Aβ40 and Aβ42 concentrations declined during the day, as did GFAP, perhaps due to more efficient CSF-to-blood clearance, which we previously provided evidence for[14]. This is supported by the present observation that plasma Aβ40 and Aβ42 concentrations increased with increasing time-to-maximum, indicative of slower CSF-to-blood clearance capacity. Furthermore, the different profiles of the various disease categories concerning the association between plasma biomarker concentrations and CSF-to-blood clearance variables suggest variation across individuals and disease types. These results point to the need for individualized assessments, i.e. individual precision diagnostics. This point was illustrated when comparing plasma biomarker concentrations between disease cohorts. As such, the iNPH and REF cohorts demonstrated differences in plasma concentrations of Aβ40, T-tau, GFAP, and NfL, which remained significant for GFAP and NfL after correction for kidney function (GFR). We cannot exclude that age difference is a confounding factor for differences between REF and iNPH subjects, but iNPH is a disease of the aged population.

In this study, we found no correlation between subjective sleep quality (assessed by the PSQI) and plasma biomarker concentrations, which was somewhat unexpected given the previous observations of different glymphatic enrichment of tracer in poor and good sleepers[18]. Despite these negative results, there is accumulating evidence for the role of sleep on biomarker concentrations in CSF and plasma. In subjects with mild-moderate AD, lack of deep sleep was associated with higher NfL concentration in CSF[68]. Furthermore, impaired sleep quality was also correlated with reduced Aβ42/Aβ40 and increased T-tau/Aβ42 ratios[69].

It might be considered a limitation of this study that lumbar CSF concentrations were not measured. On the other hand, multi-hour repeated sampling of lumbar CSF would not be feasible, as it would require a lumbar drainage system, and heavily interfere with the interpretation of the CSF tracer studies. We also decided not to retrieve lumbar CSF prior to the intrathecal administration of gadobutrol in order not to affect the normal CSF circulation. In future dedicated experimental setups, simultaneous sampling of CSF and plasma over time should nevertheless be considered. Another limitation is that the intrathecal use of gadobutrol is currently off-label. For this reason, we will explore in future studies the utility of a computer tomography (CT) contrast agent, which is approved for intrathecal use. As MRI and CT contrast agents share many of the same properties, such as similar molecular size, being hydrophilic and inert, we hypothesize that these groups of contrast agents also share the same clearance pathways and clearance rates.

In conclusion, plasma concentrations of neurodegeneration biomarkers are associated with indices of glymphatic and meningeal lymphatic functions. The plasma concentrations also varied during the daytime, supporting previous data that CSF-mediated molecular clearance is under circadian control[9]. Overall, solute clearance from CSF may be a significant contributor to the plasma concentrations of neurodegeneration biomarkers, and assessment of CSF clearance capacity by administration of an exogenous tracer substance provides dynamic information that goes beyond what can be retrieved by point measurements of endogenous solutes in CSF and blood. The finding that CSF-to-blood clearance variables derived from pharmacokinetic

modeling[14] associated with plasma biomarker concentrations in a biomarker- and disease-specific manner, may suggest the need for individualized assessment of CSF-to-blood clearance at an individual level. In particular, this may be utilized in the preclinical phase and as a tool to monitor treatments and interventions initiated to modify CSF-mediated solute clearance.

## Methods

### Ethical permissions
The research study was approved by The Institutional Review Board (2015/1868), Regional Ethics Committee (2015/96), and the National Medicines Agency (15/04932-7), and registered in Oslo University Hospital Research Registry (ePhorte 2015/1868). Patients were included after written and oral informed consent.

### Experimental design
The study design was prospective and observational: Randomization of patients was not relevant; neither was a priori sample size calculation needed.

### Patients
Consecutive patients referred to the Department of Neurosurgery, Oslo University Hospital–Rikshospitalet, Oslo, Norway, for various tentative CSF disturbances were included. The indication for intrathecal contrast-enhanced MRI was made for clinical reasons.

Concerning the categorization of CSF disorders, reference subjects (REF) are individuals in whom no apparent evidence of CSF disturbance was identified and no indication for surgery. The hydrocephalus category was either subjects with idiopathic normal pressure hydrocephalus (iNPH) who underwent shunting with a demonstrated clinical improvement, thereby qualifying for the diagnosis of Definite iNPH according to the Japanese guidelines[19] or subjects with communicating hydrocephalus (cHC) not defined as iNPH. The subjects with arachnoid cysts (AC) underwent surgery followed by post-operative clinical improvement. The idiopathic intracranial hypertension (IIH) subjects were shunted with clinical improvement thereafter. The patients with spontaneous intracranial hypotension (SIH) had an identified CSF leakage, requiring surgery to close the leakage.

### Assessment of tracer enrichment in CSF and extravascular brain
The MRI contrast agent gadobutrol was used as a CSF tracer and was administered intrathecally in doses of either 0.25 or 0.5 mmol (0.25 or 0.5 ml of 1.0 mmol/ml gadobutrol; Gadovist, Bayer Pharma AG, Berlin, Germany). Standardized T1-weighted MRI scanning was performed before and at defined time points after administration of gadobutrol, utilizing a 3 Tesla Philips Ingenia MRI scanner (Philips Medical Systems, Best, The Netherlands) was used for MRI. Equal imaging protocol settings were applied at all time points to acquire sagittal 3D T1-weighted volume scans, with the following imaging parameters: repetition time = "shortest" (typically 5.1 ms), echo time = "shortest" (typically 2.3 ms), Flip angle = 8 degrees, a field of view = 256 × 256 cm and matrix = 256 × 256 pixels (reconstructed 512 × 512). Hundred and eighty-four over-contiguous (overlapping) slices with 1 mm thickness were sampled, which were automatically reconstructed to 368 slices with 0.5 mm thickness. The duration of each image acquisition was 6 min and 29 s. To secure consistency and reproducibility of the MRI slice placement and orientation, slice orientation of image stacks were defined using an automated anatomy recognition protocol based on landmark detection in MRI data (SmartExam™, Philips Medical Systems, Best, The Netherlands) for every time point. Gadobutrol increases the T1 relaxation of water, which provides a higher T1 signal intensity at the image gray scale, thereby providing a semi-quantitative measure of the tracer level. The images were post-processed using FreeSurfer software (version 6.0) to determine the percentage change in normalized T1 signal

units, indicative of tracer enrichment. For this, FreeSurfer software (version 6.0) (http://surfer.nmr.mgh.harvard.edu/) was used for segmentation, parcellation, and registration/alignment of the longitudinal data, and to determine the increase in T1 signal caused by the CSF tracer, as previously reported[70]. We further adjusted for changes in the gray scale between MRI scans, by dividing the T1 signal unit for each time point by the T1 signal unit of a reference region of interest (ROI) for the respective time point. The reference ROI was placed within the posterior part of the orbit, as previously described[17]. The ratio is the *normalized T1 signal unit*s, which corrects for baseline changes of image gray scale due to automatic image scaling. Tracer enrichment was semi-quantified as the percentage change in normalized T1 signal at different time points, relative to the pre-contrast injection.

### Plasma biomarkers
Venous blood samples were obtained at empirically determined regular time points up to about 48 h after intrathecal administration of gadobutrol at the time of the MRI acquisitions and stored in the refrigerator (4 °C) for a few hours before centrifuge and thereafter storage in an ultra-freezer (−80 °C). Aβ40, Aβ42, GFAP, and NfL concentrations were measured using the Single molecule array (Simoa) Human Neurology 4-Plex E (N4PE) assay, whilst T-tau was measured using the Simoa Tau Advantage kit according to instructions from the kit manufacturer (Quanterix, Billerica MA). All measurements were performed on an HD-X instrument (Quanterix, Billerica MA) in one round of experiments using one batch of reagents by board-certified laboratory technicians who were blinded to the clinical data. Intra-assay coefficients of variation were below 10%.

### Assessment of CSF-to-blood clearance capacity
The CSF-to-blood clearance capacity was estimated by quantifying the intrathecally administered gadolinium in plasma according to a previously described method[13,14]. Measured gadolinium concentrations were recalculated to gadobutrol concentrations. We have previously developed a population pharmacokinetic model to determine individual pharmacokinetic parameters of intrathecally administered gadobutrol[14]. In short, a two-compartment model with first-order elimination from the central compartment was developed using a non-parametric adaptive grid approach implemented in Pmetrics for R. Posterior individual parameter values and posterior individually predicted concentrations obtained from the final population pharmacokinetic model run with the complete dataset were used for all pharmacokinetic calculations. Predictions were made in 1-min intervals from the time of administration, up to 72 h. The following pharmacokinetic variables were included: (i) The absorption half-life ($T_{1/2, abs}$), which is the time for half the amount of gadobutrol in the CSF to be cleared to blood, serving as one surrogate marker for CSF-to-blood clearance capacity of gadobutrol. (ii) Time to maximum concentration ($T_{max}$) in plasma, which was obtained directly from the individual predictions. (iii) Lag-time of absorption to blood ($T_{lag}$) is the model-estimated time for the tracer to reach the site of clearance in CSF, implying that longer $T_{lag}$ means that the tracer stays longer within CSF or that it takes longer before the clearance process to blood starts. (iv) Time to maximum concentration ($C_{max}$) in plasma were obtained directly from the individual predictions. (v) Area under the concentration-time curve from zero to infinity ($AUC_{0-\infty}$) is a measure of systemic exposure to gadobutrol. The $C_{max}$ and $AUC_{0-\infty}$ were normalized with respect to dose in order to compare parameters across multiple doses. We have previously demonstrated dose linearity in the relevant range[14].

### Assessment of sleep disturbance
We assessed subjective sleep quality utilizing the Pittsburgh Sleep Quality Index (PSQI) questionnaire[71], referring to the patient's sleep

over the past months, *i.e.* how their sleep is in general, not exclusively at the time of the MRI/blood sampling. A global PSQI score ≤5 is considered indicative of good sleep quality[71].

## Statistical analyses

Statistical analyses were performed using Stata/SE 17.0 (StataCorp, College Station, TX). Statistical significance was accepted at the 0.05 level (two-tailed). Daytime variation in biomarker concentrations was analyzed using a nonlinear model; a fractional polynomial linear regression with a maximum of one degree of the fractional polynomial and robust standard error for repeated measurements of the same subject. The plots were presented with the linear prediction (estimated mean from the regression model) and 95% confidence interval. A general linear model or a two-sample *t*-test assessed the mean difference between groups. Due to skewed T-tau values, we used the logarithm of T-tau values ($\log_{10}$T-tau). A few unusually high values of GFAP above 3000 pg/ml and NfL above 400 pg/ml were assumed to be outliers and removed from the dataset.

## Reporting summary

Further information on research design is available in the Nature Portfolio Reporting Summary linked to this article.

## Data availability

The data presented in this work is available upon request. Source data are provided with this paper.

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

## Acknowledgements

The authors thank Dr. Øivind Gjertsen, Dr. Bård Nedregaard, and Dr. Ruth Sletteberg from the Department of Radiology, Oslo University Hospital—Rikshospitalet, who performed the intrathecal gadobutrol injections in all study subjects. We also sincerely thank the Intervention Centre and Department of neurosurgery at Oslo University Hospital Rikshospitalet for providing valuable support with MR scanning and care-taking of all study subjects throughout the study. L.M.V. is supported by grants from Health South-East, Norway (grant 2020068). K.B. is supported by the Swedish Research Council (#2017-00915 and #2022-00732), the Alzheimer Drug Discovery Foundation (ADDF), USA (#RDAPB-201809-2016615), the Swedish Alzheimer Foundation (#AF-930351, #AF-939721, and #AF-968270), Hjärnfonden, Sweden (#FO2017-0243 and #ALZ2022-0006), the Swedish state under the agreement between the Swedish government and the County Councils, the ALF-agreement (#ALFGBG-715986 and #ALFGBG-965240), the European Union Joint Program for Neurodegenerative Disorders (JPND2019-466-236), the National Institute of Health (NIH), USA, (grant #1R01AG068398-01), the Alzheimer's Association 2021 Zenith Award (ZEN-21-848495), and the Alzheimer's Association 2022-2025 Grant (SG-23-1038904 QC). H.Z. is a Wallenberg Scholar supported by grants from the Swedish Research Council (#2018-02532), the European Union's Horizon Europe research and innovation program under grant agreement No 101053962, Swedish State Support for Clinical Research (#ALFGBG-71320), the Alzheimer Drug Discovery Foundation (ADDF), USA (#201809-2016862), the AD Strategic Fund and the Alzheimer's Association (#ADSF-21-831376-C, #ADSF-21-831381-C, and #ADSF-21-831377-C), the Bluefield Project, the Olav Thon Foundation, the Erling-Persson Family Foundation, Stiftelsen för Gamla

Tjänarinnor, Hjärnfonden, Sweden (#FO2022-0270), the European Union's Horizon 2020 research and innovation program under the Marie Skłodowska-Curie grant agreement No 860197 (MIRIADE), the European Union Joint Program—Neurodegenerative Disease Research (JPND2021-00694), and the UK Dementia Research Institute at UCL (UKDRI-1003).

## Author contributions

Conceptualization and design, P.K.E., G.R., K.B., and H.Z.; Blood sampling, P.K.E. and A.L. Plasma biomarker analysis: K.B. and H.Z. Statistical analysis: P.K.E. and A.P. FreeSurfer analysis: L.M.V. Population pharmacokinetic modeling: M.H.H. Writing—original draft: P.K.E. Writing—review and editing, P.K.E., A.L., A.P., L.M.V., M.H.H., G.R., K.B., and H.Z. All authors approved the final manuscript.

## Competing interests

P.K.E. and G.R. are shareholders in BrainWideSolutions AS, Oslo, Norway, which is a holder of patent US 11,272,841. K.B. has served as a consultant on advisory boards or on data monitoring committees for Abcam, Axon, BioArctic, Biogen, and JOMDD/Shimadzu. Julius Clinical, Lilly, MagQu, Novartis, Ono Pharma, Pharmatrophix, Prothena, Roche Diagnostics, and Siemens Healthineers, and is a co-founder of Brain Biomarker Solutions in Gothenburg AB (BBS), which is a part of the GU Ventures Incubator Program, outside the work presented in this paper. H.Z. has served on scientific advisory boards and/or as a consultant for Abbvie, Acumen, Alector, ALZPath, Annexon, Apellis, Artery Therapeutics, AZTherapies, CogRx, Denali, Eisai, Nervgen, Novo Nordisk, Passage Bio, Pinteon Therapeutics, Red Abbey Labs, reMYND, Roche, Samumed, Siemens Healthineers, Triplet Therapeutics, and Wave, has given lectures in symposia sponsored by Cellectricon, Fujirebio, Alzecure, Biogen, and Roche, and is a co-founder of Brain Biomarker Solutions in Gothenburg AB (BBS), which is a part of the GU Ventures Incubator Program (outside submitted work). The remaining authors declare no competing interests.
