## [Peer Review File · Nature Communications]

REVIEWER COMMENTS

Reviewer #1 (Remarks to the Author):

This study examined 106 individuals with both intrathecal injected Gadobutrol MRI exam and plasma biomarkers test. This unique data set is extremely valuable and will help us better understand the brain CSF clearance. However, a more thorough discussion is needed for the glymphatic and CSF-to-blood clearance.

First, CSF Tracers, the pia mater permeability and the paravascular pathways.

1) the 24-48 hours clearance phase for Gadobutrol. Different CSF tracers showed different CSF clearance speed. Comparing to the 24-48 hours clearance phase for Gadobutrol, the translocation time from the CSF to the brain parenchyma is much shorter for several radiotracers in both the human study[PMID: 32355870] and the animal study[PMID: 22896675]. Why Gadobutrol takes so long to get into brain from the CSF? The perivascular space (PVS) is covered by a thin sheath of pia mater and indirectly communicating with the subarachnoid space. [PMID: 2254158] The sheath of pia mater is a thin fibrous tissue that is permeable to water and small solutes. I'm wondering if the permeability for different CSF tracers may affect the clearance speed.

2)line 350: A β clearance pathway. some evidence supports the CSF clearance pathway of Ab. such as the PVS burden correlate with brain Ab pathology, and a recent paper showed that Amyloid-beta is present in human lymph nodes and greatly enriched in those of the cervical region. [PMID: 34057798] The study should cite evidence from both sides and make a balanced discussion for Ab clearance.

Second, CSF-to-blood clearance (proxy of meningeal lymphatic egress), CSF circulation and Arachnoid granulations.

line 117, 327: the CSF-to-blood clearance may reflect the total meningeal lymphatic clearance capacity. This conclusion didn't consider the arachnoid granulation is the major pathway for CSF circulation. The arachnoid granulation in superior sinus is the major pathway of CSF absorption. Is the MRI tracer in CSF absorbed by arachnoid granulation? Comparing to arachnoid granulation, what type of molecular and how much is cleared through the meningeal lymphatic vessel? if unclear, how to conclude that the tracer was cleared through the meningeal lymph, not through the conventional CSF circulation pathway? A detailed discussion needed. Following the last question, is CSF biomarkers, such as tau, Ab, GFAP and NfL absorbed by the arachnoid granulation?

line 141 and 267: tau lacks a dedicated BBB-transporter. recent study (PMID: 27662303) showed that tau can pass BBB. the paper need rethink of the tau clearance pathway.

Reviewer #2 (Remarks to the Author):

In this manuscript, the authors wish to investigate the association between CSF clearance and neurodegenerative plasma biomarker behavior. They rely on data collected in a cohort of consecutive patients with an indication for intrathecal MRI with contrast. Therefore, the study cohort is composed of many patients with severely disturbed CSF clearance.

The main conclusion of this very complex study, which requires a number of explanations and assumptions to interpret the data, is firstly that “ solute clearance from CSF seems to be a significant contributor to the plasma concentrations of neurodegeneration biomarkers”. This conclusion is only indirect and is based on the correlation between the MRI data and the blood measurements. Its evidence based value is therefore low. Moreover, the relationship between CSF and blood concentrations of biomarkers of neurodegeneration, is well established by numerous studies. Sophisticated measurements after stable isotope labeling have, for example, established this relationship for amyloid peptides (see <https://pubmed.ncbi.nlm.nih.gov/28734653/> <https://pubmed.ncbi.nlm.nih.gov/35087179/>). Parallel measurements in CSF and blood made in many studies for the different biomarkers also lead to this conclusion.

The second main conclusion of the work (see the last paragraph of the discussion), “suggest the need for individualized assessment of CSF-to-blood clearance at individual level.” This conclusion is rather a consequence of the heterogeneity of the cohort, its poor characterization (lack of knowledge of the presence of amyloidopathy, neuronal damage...) and the variability of the method used in the study.

In addition to the fact that the study does not provide very new and indisputable data on the transfer of biomarkers from CSF to blood, it raised several problems and questions listed below.

- The data were obtained in a very heterogeneous cohort which in fact does not contain patients with neurodegenerative disease. The authors thus overinterpret their data by extending their observations to neurodegenerative diseases.

- While it is true that the iNPH group may resemble the prodromal forms of Alzheimer's disease to some extent, only 50% of these patients are known to have amyloidopathy. In the absence of CSF

data or amyloid or tau imaging, one must conclude that the population is very poorly characterized and it is too risky to consider it a homogeneous entity, even within the groups defined in the study.

- The issue of population characteristics is also raised by the blood biomarker results. If the REF group is considered a "non-pathological situation", the level of biomarkers in the other groups is often high, as much or more than in Alzheimer's. Some of them show unexpectedly high levels suggesting neuronal damage. This means that the population cannot be used to model a physiological or neurodegenerative situation.

- The fact that the study relies heavily on the results obtained with total tau (t-tau) is in this context very revealing of the non-adaptation of the cohort to neurodegenerative diseases. It is indeed well known that the level of t-tau is a very poor indicator of these diseases. One of the reasons for this is that the detection of t-tau in the blood includes tau that does not originate in the brain. Seeing a significant variation in tau means that a very large variation in tau exists in the CSF, related to an abnormal BBB, brain damage, ischemia, etc

- The age difference between the groups, especially between REF and iNPH, hinders the interpretation of the data with respect to blood biomarker levels in particular.

- In the study, A β 40 and A β 42 often differ from other biomarkers (not affected by impaired tracer clearance, no daytime change). These results are not in line with the literature and the explanation given for The Hence, for the entire

- The absence of CSF measurement not only hinders the knowledge of the pathological situation of the cohort but also the interpretation of the data. Indeed, as discussed in the manuscript, the presence of biomarkers in the blood may result from a transit from the brain parenchyma to the blood without transit through the CSF. The absence of this information therefore makes the interpretation of the data uncertain.

- An explanation and discussion of the differences observed between the groups (in Tables 1, 2..) is necessary. Indeed, the differences between the groups are significant and do not go in the same direction at 24 or 48 hours. For example, iNPHs have similar CSF values at 24 hours and much higher values at 48 hours. In the cerebral cortex the situation is very different and so on...

- Nycthemeral variation in blood biomarker concentration is important information, but it also depends on patients' sleep patterns. Subjective sleep quality (PSQI) mentioned in the supplementary data, but not discussed at all in the text (!!) is probably not sufficient to interpret the data.

- The figures lack numeration

Manuscript - NCOMMS-22-47605-R1 - Response to reviewers with reference to manuscript with changes highlighted

Reviewer #1:

General comment #1:

This study examined 106 individuals with both intrathecal injected Gadobutrol MRI exam and plasma biomarkers test. This unique data set is extremely valuable and will help us better understand the brain CSF clearance. However, a more thorough discussion is needed for the lymphatic and CSF-to-blood clearance.

Answer: We greatly appreciate the reviewers interest in our work and have expanded the discussion, as suggested.

Specific comment #1:

First, CSF Tracers, the pia mater permeability and the paravascular pathways.

1) the 24-48 hours clearance phase for Gadobutrol. Different CSF tracers showed different CSF clearance speed. Comparing to the 24-48 hours clearance phase for Gadobutrol, the translocation time from the CSF to the brain parenchyma is much shorter for several radiotracers in both the human study[PMID: 32355870] and the animal study[PMID: 22896675]. Why Gadobutrol takes so long to get into brain from the CSF? The perivascular space (PVS) is covered by a thin sheath of pia mater and indirectly communicating with the subarachnoid space. [PMID: 2254158] The sheath of pia mater is a thin fibrous tissue that is permeable to water and small solutes. I'm wondering if the permeability for different CSF tracers may affect the clearance speed.

Answer: This is a timely and highly relevant question, also highlighted by a very recent report about a subarachnoid lymphatic-like membrane (mesothelium) dividing the subarachnoid space into functional compartments (Møllgård et al, Science, 2023). The authors of this report concluded that solutes >3 kDa did not pass this membrane. The 1990 paper referred to by the reviewer [PMID 2254158] is highly relevant in this regard. Marked differences between rodents and humans regarding speed of tracer propagation in subarachnoid spaces and brain are well documented (e.g. refs 12 and 16). However, given that the plasma biomarkers referred to in our study have different molecular sizes (and definitely larger than our CSF tracer), variable passage to the subarachnoid spaces, and meningeal lymphatic structures could be one factor behind the different behaviors of the plasma biomarkers, which we have commented on now (page 9, para 1). Exact answers to this question are, however, not available at this stage. The comment about differences from radiotracers [PMID 32355870] (where images were obtained 8 and 24 hours after intrathecal ^{99m}Tc-DPTA) is complicated by the fact that the tracer half-life is about 6 hours (as already commented on with reference to this particular work (page 9, para 2), which limits its assessment after 24 hours. However, several aspects of the results of this radiotracer study are comparable with the presently reported data. The discussion about role of permeability of biomarkers has been included (page 9, para 1), and the proposed references are in the paper. With regard to the other study [22896675], two-photon imaging at the cortical surface (about 0.2 mm below surface) was done in mice, while we here examined tracer enrichment in the entire brain of humans. Species differences contribute to the observed differences in tracer enrichment. These studies and the present are not in conflict but rather complimentary.

Specific comment #2:

2)line 350: Aβ clearance pathway. some evidence supports the CSF clearance pathway of Ab. such as the PVS burden correlate with brain Ab pathology, and a recent paper showed that Amyloid-beta is present in human lymph nodes and greatly enriched in those of the cervical region. [PMID: 34057798] The study should cite evidence from both sides and make a balanced discussion for Ab clearance.

Answer: We agree that this is an important study, and have referred to it now in addition to the other efflux routes (via BBB and cellular degradation) (page 8, para 2).

Specific comment #3:

Second, CSF-to-blood clearance (proxy of meningeal lymphatic egress), CSF circulation and Arachnoid granulations.

line 117, 327: the CSF-to-blood clearance may reflect the total meningeal lymphatic clearance capacity. This conclusion didn't consider the arachnoid granulation is the major pathway for CSF circulation. The arachnoid granulation in superior sinus is the major pathway of CSF absorption. Is the MRI tracer in CSF absorbed by arachnoid granulation? Comparing to arachnoid granulation, what type of molecular and how much is cleared through the meningeal lymphatic vessel? if unclear, how to conclude that the tracer was cleared through the meningeal lymph, not through the conventional CSF circulation pathway? A detailed discussion needed. Following the last question, is CSF biomarkers, such as tau, Ab, GFAP and NfL absorbed by the arachnoid granulation?

Answer: First, for years the role of the arachnoid granulations in CSF efflux has been disputed. With regard to the CSF-to-blood clearance variables retrieved from the pharmacokinetic model, it may be considered an advantage that the interpretation of these variables do not depend on the role of arachnoid granulations as clearance route. Therefore, we wrote this in the first version: *"It is important to note that this method measures total CSF-to-blood clearance capacity, independent of clearance route. For substances also passing the BBB, direct passage to blood may contribute. In subjects with impaired BBB integrity, which must be anticipated in older subjects⁴⁶, passage of substances directly to blood may be expected to some degree. In other rare diseases, e.g. spontaneous intracranial hypotension due to CSF-to-venous-fistula⁴⁷, there is abnormal direct passage to blood. [...]. However, knowledge of the exact clearance route is not critical, as the CSF-to-blood clearance aims to measure total clearance capacity, independent of clearance pathway. For compounds largely excreted via CSF, the CSF-to-blood clearance capacity may largely reflect the total meningeal lymphatic clearance capacity."* We now also have referred to our previous study (Ringstad, Eide, Nat Comm, 2020) that the CSF tracer strongly enriched parasagittal dura but only were seen in a few arachnoid granulations, suggesting that the volume fraction of efflux via arachnoid granulations is minor (page 9, last para). Our interpretation that the CSF-to-blood clearance variables may indicate meningeal lymphatic clearance capacity primarily relies on our CSF tracer studies on enrichment of parasagittal dura.

Specific comment #4:

line 141 and 267: tau lacks a dedicated BBB-transporter. recent study (PMID: 27662303) showed that tau can pass BBB. the paper need rethink of the tau clearance pathway.

Answer: We thank the reviewer for making us aware of this study and have included the work, and it has been included (page 8, para 2).

Reviewer #2:

General comment #1:

In this manuscript, the authors wish to investigate the association between CSF clearance and neurodegenerative plasma biomarker behavior. They rely on data collected in a cohort of consecutive patients with an indication for intrathecal MRI with contrast. Therefore, the study cohort is composed of many patients with severely disturbed CSF clearance.

Answer: We thank the referee for thorough review of our work, though some of the comments below may indicate possible misinterpretations. We did not only address association between CSF clearance and plasma biomarkers, but association of plasma biomarkers with indices of both glymphatic and meningeal lymphatic functions. Further, the statement that patients had "severely

disturbed CSF clearance” is a stretch (see response to specific comment #12). Our response to the specific critics by the reviewer is addressed below.

Specific comment #1:

The main conclusion of this very complex study, which requires a number of explanations and assumptions to interpret the data, is firstly that “solute clearance from CSF seems to be a significant contributor to the plasma concentrations of neurodegeneration biomarkers”. This conclusion is only indirect and is based on the correlation between the MRI data and the blood measurements. Its evidence based value is therefore low.

Answer: We agree that the wording of the conclusion could be improved, and have therefore modified the conclusion (Abstract; page 12, para 3). We also have clarified better the aim of study (page 3, last para; page 4, first para).

Specific comment #2:

Moreover, the relationship between CSF and blood concentrations of biomarkers of neurodegeneration, is well established by numerous studies. Sophisticated measurements after stable isotope labeling have, for example, established this relationship for amyloid peptides (see <https://pubmed.ncbi.nlm.nih.gov/28734653/> / <https://pubmed.ncbi.nlm.nih.gov/35087179/>). Parallel measurements in CSF and blood made in many studies for the different biomarkers also lead to this conclusion.

Answer: The research question of the present study was not the relationship between CSF and blood concentrations of biomarkers. Our focus was how plasma concentrations of biomarkers associate with indices of glymphatic (examined by enrichment in brain of an intrathecal CSF tracer) and meningeal lymphatic functions (examined by pharmacokinetic modeling of CSF-to-blood clearance). To avoid misinterpretation, this has been made clearer (title changed, and Introduction modified; page 3, last para; page 4, first para). In this regard, it is important to be aware that the CSF-to-blood clearance variables revealed by pharmacokinetic modeling of an intrathecal tracer provide another perspective than parallel measurements of CSF and blood concentrations (e.g. amyloid- β or tau). Amendments were made to better describe the intention and output of the study.

Specific comment #3:

The second main conclusion of the work (see the last paragraph of the discussion), “suggest the need for individualized assessment of CSF-to-blood clearance at individual level.” This conclusion is rather a consequence of the heterogeneity of the cohort, its poor characterization (lack of knowledge of the presence of amyloidopathy, neuronal damage...) and the variability of the method used in the study.

Answer: Our statement is based on the observations of highly inter-individual differences in CSF-to-blood clearance variables, both within and across patient groups (Figure 3; Table 2). This is a most significant observation, demonstrating large inter-individual variation in CSF-to-blood clearance capacity. The conclusion is therefore justified. Given the Reviewer’s concerns about heterogeneity between patients, Table 1 now shows demographic information about the individual groups. The statement that the patient groups are heterogeneous and poorly characterized is not justified; they fulfilled the criteria of their respective diagnoses. It may rather be considered an advantage to include different patient groups.

Specific comment #4:

In addition to the fact that the study does not provide very new and indisputable data on the transfer of biomarkers from CSF to blood, it raised several problems and questions listed below.

Answer: In this work, we for the first time address associations between plasma concentrations of biomarkers and indices of glymphatic function, examined by the current gold-standard in vivo method, namely intrathecal contrast-enhanced MRI. In addition, we for the first time show the

association between plasma biomarker concentrations and novel CSF-to-blood variables based on a pharmacokinetic model. We here present novel data previously not reported in the literature, and therefore provide new perspectives, e.g. in comparison to previous reports about parallel measurements of CSF and blood concentrations of biomarkers. The latter approach has methodological weaknesses since CSF catheters and repeated CSF taps may affect the measured biomarker concentrations. Parallel CSF/blood concentrations do neither tell about the underlying transport mechanisms.

Specific comment #5:

- The data were obtained in a very heterogeneous cohort which in fact does not contain patients with neurodegenerative disease. The authors thus overinterpret their data by extending their observations to neurodegenerative diseases.

Answer: We agree with the reviewer that we should not over-interpret how altered plasma concentrations predict neurodegeneration. We therefore have to some extent modified the discussion to take into account this critique (page 7, para 3; page 10, last para). It is, however, well established that the iNPH disease is a neurodegenerative disease. The present iNPH patients were Definite iNPH according to the Japanese guidelines (page 4, para 2; page 13, para 2). Moreover, it has been made clear that the primary intention of this study was on a general basis to examine the association between plasma concentrations of neurodegeneration biomarkers and indices of glymphatic and meningeal lymphatic function (page 3, last para). In this context, inclusion of different categories may rather be considered an advantage, as glymphatic and/or meningeal lymphatic clearance failure should be a general mechanism behind changes in plasma biomarker concentrations. Furthermore, there is in neuroscience a profound interest in diagnosing neurodegeneration in the pre-symptomatic stage. Research on neurodegeneration biomarkers should therefore not be limited to assessments in patients with established dementia disease only.

Specific comment #6:

- While it is true that the iNPH group may resemble the prodromal forms of Alzheimer's disease to some extent, only 50% of these patients are known to have amyloidopathy. In the absence of CSF data or amyloid or tau imaging, one must conclude that the population is very poorly characterized and it is too risky to consider it a homogeneous entity, even within the groups defined in the study.

Answer: See, our comment to Specific comment #5. The iNPH population is not poorly characterized as they fulfilled the criteria of "Definite" iNPH according to the Japanese guidelines (page 4, para 2; page 13, para 2). We further would disagree that it is essential which proportion of the present Definite iNPH patients who presented with amyloidopathy. Our research question was not which plasma biomarker levels that represent risk of developing amyloidopathy.

Glymphatic/meningeal clearance failure may theoretically be present years before manifestation of amyloidopathy. We agree that CSF concentration might add value, but repeated CSF drainage via an indwelling catheter would impair or even destroy the interpretation of the tracer enrichment in CSF and brain. Moreover, from an ethical perspective, repeated CSF taps via a lumbar drain in addition to the repeated blood samples and MRI acquisitions would hardly be justified.

Specific comment #7:

- The issue of population characteristics is also raised by the blood biomarker results. If the REF group is considered a "non-pathological situation", the level of biomarkers in the other groups is often high, as much or more than in Alzheimer's. Some of them show unexpectedly high levels suggesting neuronal damage. This means that the population cannot be used to model a physiological or neurodegenerative situation.

Answer: The REF population consisted of patients in whom no apparent CSF disturbance was diagnosed, and does not claim to include a "non-pathological" situation. Notably, the diagnostic

procedure of intrathecal contrast enhanced MRI cannot be done in healthy subjects as the tracer (gadobutrol) is administered off-label. It is further difficult to see why these patients should not be candidates for assessing how blood biomarkers associate with glymphatic function and meningeal lymphatic function in general. The comment that biomarker concentrations are unexpectedly high suggesting neuronal damage, may rather point to the need of assessing plasma biomarkers in various patient cohorts to get a broader picture. Up to now, plasma biomarkers have largely been explored in Alzheimer's disease, frontotemporal dementia, atypical parkinsonian disorders and post-traumatic injury (page 7, para 3). We therefore see it as an advantage to assess plasma biomarkers over a range of conditions having possibly different meningeal and glymphatic clearance functions.

Specific comment #8:

- *The fact that the study relies heavily on the results obtained with total tau (t-tau) is in this context very revealing of the non-adaptation of the cohort to neurodegenerative diseases. It is indeed well known that the level of t-tau is a very poor indicator of these diseases. One of the reasons for this is that the detection of t-tau in the blood includes tau that does not originate in the brain. Seeing a significant variation in tau means that a very large variation in tau exists in the CSF, related to an abnormal BBB, brain damage, ischemia, etc*

Answer: T-tau was one of five plasma biomarkers included in the study, and as already commented on, our focus was how plasma concentrations of these neurodegeneration biomarkers associate with indices of glymphatic and meningeal lymphatic functions. While we agree that T-tau may have different origins, it would rather be expected that factors such as BBB damage, ischemia etc. would mask significant correlations, not explain the significance levels.

Specific comment #9:

- *The age difference between the groups, especially between REF and iNPH, hinders the interpretation of the data with respect to blood biomarker levels in particular.*

Answer: We fully agree that age-differences contribute to the observed differences. However, as already commented on, some differences between iNPH and REF subjects became non-significant after statistical adjustment for age (page 6, last para). However, this fact does not mean that the differences are not of significance. This is because iNPH is a disease of the elderly (typically aged >65-70 years), and therefore age per se is a significant factor. Therefore, these differences between REF and iNPH caused by ageing are of significance.

Specific comment #10:

- *In the study, A β 40 and A β 42 often differ from other biomarkers (not affected by impaired tracer clearance, no daytime change). These results are not in line with the literature and the explanation given for The Hence, for the entire*

Answer: We have commented that amyloid- β has several efflux routes (page 8, para 2), which may be a factor behind the present observations. However, since correlations between plasma concentrations of A β 40 and A β 42 and indices of glymphatic/meningeal lymphatic function have previously not been reported in the literature, it is hard to compare the present results with previous ones.

Specific comment #11:

- *The absence of CSF measurement not only hinders the knowledge of the pathological situation of the cohort but also the interpretation of the data. Indeed, as discussed in the manuscript, the presence of biomarkers in the blood may result from a transit from the brain parenchyma to the blood without transit through the CSF. The absence of this information therefore makes the interpretation of the data uncertain.*

Answer: As already commented on (see Spec comment #6), ideally it would be beneficial to include CSF concentrations. This has been commented on in the Limitations section (page 12, para 2). However, with the present study design, it would not be feasible with simultaneous CSF measurements since CSF drainage would interfere with the transport of CSF tracer, and make interpretation of CSF tracer observations impossible. The burden on the patient was another consideration. We therefore could not include parallel CSF samples in the present study design. On the other hand, the interpretation of the CSF-to-blood clearance variables from the pharmacokinetic model does not depend on CSF measurements. In addition, it is presently not known which fraction of the metabolites that are excreted via CSF, across the BBB or undergoing cellular degradation. This fact also makes the present study design of interest, namely to explore association between plasma concentrations of biomarkers and indices of glymphatic and meningeal lymphatic concentrations.

Specific comment #12:

- *An explanation and discussion of the differences observed between the groups (in Tables 1, 2..) is necessary. Indeed, the differences between the groups are significant and do not go in the same direction at 24 or 48 hours. For example, iNHPs have similar CSF values at 24 hours and much higher values at 48 hours. In the cerebral cortex the situation is very different and so on...*

Answer: This comment is hard to understand, it seems as the reviewer has misinterpreted Tables 1-2 since the tables do not show what the reviewer claim: *“Indeed, the differences between the groups are significant and do not go in the same direction at 24 or 48 hours. For example, iNHPs have similar CSF values at 24 hours and much higher values at 48 hours.”* However, it is the association between indices of glymphatic and meningeal lymphatic markers and plasma concentrations that are central for the present data, and not differences between groups per se. As commented on, given the focus on heterogeneity between patient groups, Table 1 now presents information about the patient groups included in the study. It may as well be commented that the comparable results between groups regarding CSF-to-blood clearance (Table 2) argue against the reviewer’s general statement that these cohorts had *“severely disturbed CSF clearance”*.

Specific comment #13:

- *Nycthemeral variation in blood biomarker concentration is important information, but it also depends on patients' sleep patterns. Subjective sleep quality (PSQI) mentioned in the supplementary data, but not discussed at all in the text (!!) is probably not sufficient to interpret the data.*

Answer: We thank the reviewer for this comment, and agree that sleep quality could impact plasma biomarker concentrations. Therefore, in the revised version we have included results about subjective sleep quality, which also has now been discussed (page 6, para 4; page 11, last para; page 14, last para). The new results have been presented in Supplementary Table 1 and Supplementary Fig. 4, and show that the present results cannot be explained by differences in subjective sleep quality between groups.

Specific comment #14:

- *The figures lack numeration*

Answer: This was probably related to an error during the file upload process, has been corrected.

REVIEWERS' COMMENTS

Reviewer #1 (Remarks to the Author):

The revised manuscript has been much improved. Before accepting it, I have a few more questions want to discuss with authors.

First, this paper raised an important hypothesis of brain CSF clearance and plasma biomarkers based on observations of a clinical dataset. But more study is needed before getting a final conclusion. The author should lower the tone of their claims wording. For example, Line 50 in abstract, change “The results provide in vivo human evidence that” to “The results suggest that”.

Second. Glymphatic clearance is an emerging research field, we should open to all different type of tracers to study the CSF clearance. A very recent paper by Dr. Nedergaard [PMID: 36809165] showed radiotracer is a promising tool for imaging the glymphatic system and showed a greater range of tracer concentrations in CSF than MRI tracer. I would suggest removing the “gold standard” for the MRI CSF tracer in the Line 310. And in Line 345, the discussion of radiotracer half-life is misleading. Radiotracer can be used to dynamic scan which is 3-4 times longer than their half-life. the DTPA can be detected within 24 hours. Please remove incorrect critics.

Third, the author cited their previous publication to support the argument that the CSF tracer strongly enriched parasagittal dura but only were seen in a few arachnoid granulations, suggesting that the volume fraction of efflux via arachnoid granulations is minor (page 9, last para). The tracer enrichment is a good indicator to identify the CSF efflux pathway but can't claim the volume fraction of efflux or the flow function. The tracer enrichment in those pathways, in another way, can be interpreted as tracer residual (flow stasis) , If the tracer accumulated in arachnoid granulations is cleared rapidly to the blood, the tracer concentration may remain lower. On the other hand, the human CSF circulation 3-4 times a day, does it too slow if mainly depends on the meningeal lymphatic drainage? These interesting discoveries give us another thought of the CSF-blood clearance pathway, but more studies are needed before reaching a conclusion.

Reviewer #2 (Remarks to the Author):

We thank the authors for the revised version of the manuscript which responds to the commentary and adds and provides clearer information on certain points.

Manuscript - NCOMMS-22-47605A - Response to reviewers with reference to manuscript with changes highlighted

Reviewer #1:

General comment #1:

The revised manuscript has been much improved. Before accepting it, I have a few more questions want to discuss with authors.

Answer: We would like to thank the reviewer for a thorough review and helpful advices that have improved our work.

Specific comment #1:

First, this paper raised an important hypothesis of brain CSF clearance and plasma biomarkers based on observations of a clinical dataset. But more study is needed before getting a final conclusion. The author should lower the tone of their claims wording. For example, Line 50 in abstract, change "The results provide in vivo human evidence that" to "The results suggest that".

Answer: We agree with the reviewer and have modified the Abstract accordingly.

Specific comment #2:

Second. Glymphatic clearance is an emerging research field, we should open to all different type of tracers to study the CSF clearance. A very recent paper by Dr. Nedergaard [PMID: 36809165] showed radiotracer is a promising tool for imaging the glymphatic system and showed a greater range of tracer concentrations in CSF than MRI tracer. I would suggest removing the "gold standard" for the MRI CSF tracer in the Line 310. And in Line 345, the discussion of radiotracer half-life is misleading. Radiotracer can be used to dynamic scan which is 3-4 times longer than their half-life. the DTPA can be detected within 24 hours. Please remove incorrect critics.

Answer: We thank the reviewer for this important comment and fully agree about the value of different kinds of tracers to study glymphatic function. The study referred to is an important work and has been added. Our comments about radiotracers have been modified, as suggested by the reviewer (page 8, para 2; page 9, para 2).

Specific comment #3:

Third, the author cited their previous publication to support the argument that the CSF tracer strongly enriched parasagittal dura but only were seen in a few arachnoid granulations, suggesting that the volume fraction of efflux via arachnoid granulations is minor (page 9, last para). The tracer enrichment is a good indicator to identify the CSF efflux pathway but can't claim the volume fraction of efflux or the flow function. The tracer enrichment in those pathways, in another way, can be interpreted as tracer residual (flow stasis) , If the tracer accumulated in arachnoid granulations is cleared rapidly to the blood, the tracer concentration may remain lower. On the other hand, the human CSF circulation 3-4 times a day, does it too slow if mainly depends on the meningeal lymphatic drainage? These interesting discoveries give us another thought of the CSF-blood clearance pathway, but more studies are needed before reaching a conclusion.

Answer: We agree with the reviewer 's reasoning. The sentence about volume fraction cleared via arachnoid granulations has been rewritten to be more precise about the current knowledge.

Reviewer #2:

General comment #1:

We thank the authors for the revised version of the manuscript which responds to the commentary and adds and provides clearer information on certain points.

Answer: We thank the reviewer for reviewing our work and for constructive critique.